PREREGISTERED RESEARCH ARTICLE

# Wakefulness can be distinguished from general anesthesia and sleep in flies using a massive library of univariate time series analyses

Angus Leung[1]◉*, Ahmed Mahmoud[1]◉, Travis Jeans[2], Ben D. Fulcher[3],
Bruno van Swinderen[2], Naotsugu Tsuchiya[1,4,5]*

1 School of Psychological Sciences and Turner Institute for Brain and Mental Health, Monash University, Melbourne, Australia, 2 Queensland Brain Institute, The University of Queensland, Brisbane, Australia, 3 School of Physics, The University of Sydney, Camperdown, Australia, 4 Center for Information and Neural Networks (CiNet), Suita, Japan, 5 Advanced Telecommunications Research Computational Neuroscience Laboratories, Kyoto, Japan

◉ These authors contributed equally to this work.
* angus.leung@monash.edu (AL); naotsugu.tsuchiya@monash.edu (NT)

**Academic Editors:** Christopher Chambers, and Simon van Gaal, University of Amsterdam: Universiteit van Amsterdam, NETHERLANDS, KINGDOM OF THE

## Abstract

The neural mechanisms of consciousness remain elusive. Previous studies on both human and non-human animals, through manipulation of level of conscious arousal, have reported that specific time-series features correlate with level of consciousness, such as spectral power in certain frequency bands. However, such features often lack principled, theoretical justifications as to why they should be related with level of consciousness. This raises two significant issues: firstly, many other types of times-series features which could also reflect conscious level have been ignored due to researcher biases toward specific analyses; and secondly, it is unclear how to interpret identified features to understand the neural activity underlying consciousness, especially when they are identified from recordings which summate activity across large areas such as electroencephalographic recordings. To address the first concern, here we propose a new approach: in the absence of any theoretical priors, we should be maximally agnostic and treat as many known features as feasible as equally promising candidates. To apply this approach, we use *highly comparative time-series analysis* (*hctsa*), a toolbox which provides over 7,700 different univariate time-series features originating from different research fields. To address the second issue, we employ *hctsa* to high-quality neural recordings from a relatively simple brain, the fly brain (*Drosophila melanogaster*), extracting features from local field potentials during wakefulness, general anesthesia, and sleep. At Stage 1 of this registered report, we constructed a classifier for each feature, for discriminating wakefulness and anesthesia in a discovery group of flies (*N* = 13). At Stage 2, we assessed their performances on four independent groups of evaluation flies, from which recordings were made during anesthesia and sleep, and which were originally

**Data availability statement:** Pre-processed data from the discovery flies are available on Figshare: https://doi.org/10.26180/5ebe-420ae8d89 Pre-processed data from the pilot evaluation flies and blinding procedure are available on OSF: https://osf.io/bq5ry/?view_only=3789097395c1419db2a9eb615bc1effe Pre-processed data, hctsa values, and classification performances and within-fly effect direction consistencies for all the flies are also available on OSF: https://osf.io/8wvsq/?view_only=8a056d1c573b4f23a6cf6cea8b976ddb Analysis codes are available on Zenodo: https://doi.org/10.5281/zenodo.15370576

**Funding:** AL and TJ were supported by Australian Government Research Training Program (RTP) Scholarships. TJ and BvS were funded by the National Health and Medical Research Council Project Grant GNT1065715. BvS was funded by the National Health and Medical Research Council Grant GNT1164879. AL, BF, and NT were funded by the National Health and Medical Research Council Ideas Grant GNT1183280. NT was funded by TWCF0199 from Templeton World Charity Foundation, Inc., Australian Research Council Discovery Projects DP180104128 and DP180100396, and the National Health and Medical Research Council APP1183280. AL and NT were funded by Japan Society for the Promotion of Science, Grant-in-Aid for Transformative Research Areas (A) (23H04829, 23H04830). BF acknowledges support from the Australian Research Council (FT240100418). The funders had no role in the study design, data collection and analysis, decision to publish, or preparation of the manuscript.

**Competing interests:** The authors have declared that no competing interests exist.

**Abbreviations:** EEG, electroencephalographic; FDR, false discovery rate; *hctsa*, *h*ighly *c*omparative *t*ime *s*eries *a*nalysis; LFPs, local field potentials.

blinded to the data analysis team ($N = 49$). We found only 47 time-series features, applied to recordings obtained from the center of the fly brain, to also significantly classify wake from anesthesia or sleep in all 4 of these evaluation datasets. Most of these were related to autocorrelation, and they indicated that signals during wakefulness remained correlated to their past for a longer timescale than during anesthesia and sleep. Meanwhile, time-series features related to well-known potential markers of consciousness, such as those related to complexity or spectral power, failed to generalize across all the flies. However, many of these complexity and spectral features have a consistent direction of effect due to anesthesia or sleep across flies, suggesting that even slight variations in experiment setup can reduce generalizability of classifiers. These results caution the current state of frequent discoveries of new potential consciousness markers, which may not generalize across datasets, and point to autocorrelation as a class of dynamical properties which does.

## Introduction

The question of how physical mechanisms generate conscious arousal is a long-standing question in neuroscience. Understanding the mechanisms that support consciousness will have significant impacts in clinical assessment of loss of consciousness [1]. Historically, researchers have approached this question through identifying electrophysiological differences in brain recordings between differing levels of consciousness, such as wakefulness and general anesthesia. This approach has resulted in the discovery of multiple time-series features as markers of level of consciousness, including spectral power in different frequency bands [2–8] and measures of signal complexity in spontaneous recordings [9–12]. Despite these developments, performance in distinguishing levels of consciousness using such markers remains limited [13,14].

The limited performance of previously identified markers in distinguishing levels of consciousness, and failure to extend to new conditions, may be due to a lack of theoretical expectations as to what they should be. Historically, candidate markers were often found through visually contrasting electrophysiological recordings, such as electroencephalograms, obtained at varying levels of consciousness [15,16]. Though this approach has led to well-known markers of depth of anesthesia, it is limited by biases toward groups of time-series features for which differences in level of consciousness are visually clear (for a related issue in sleep research, see [17]). While newer markers have moved away from features which are visually clear, a similar problem applies, wherein researchers investigate features selected based on their own expertise of particular facets of time-series structure. Consequently, a vast range of other time-series features have been ignored as potential markers for conscious level.

One way of removing the bias inherent to selecting individual features to investigate as markers of conscious level is to be maximally agnostic about the types of

time-series properties which can map to consciousness. Then, we can systematically test and compare as many time-series features as feasible. This approach consists of two main components.

Firstly, "all" potential features of some given neurophysiological time series should be investigated. While comparison of multiple features has been done previously on well-established features [13,14], "all" features should be compared, not only those determined by or related to visual inspection or individual expertise in particular time-series features. While at first this may seem like a daunting task, this is feasible using *highly comparative time series analysis* (*hctsa*; [18]). *hctsa* is a computational framework which extracts from a given time series a massive number (>7,000) of univariate time-series features. These features are taken from a multitude of research fields, and include measures such as basic statistics of the distribution of time samples, linear correlations among timepoints, stationarity, entropy measures, among others. This library has been applied previously to find meaningful time-series features for such applications as detecting falls [19] to identifying physiological dynamics underlying neurological disorders [20].

Secondly, to avoid overfitting to a particular dataset, features should be validated on datasets independent from the original dataset from which the features were originally identified [21,22]. In particular, validating features on blinded datasets while utilizing a registered report approach can help mitigate biases toward certain features [23]. While standards are shifting toward testing proposed features in independent samples [14,24,25], "cross-validation", a method which splits data into training and testing sets is still widely utilized, likely due to the cost and clinical problems of obtaining independent datasets. This is especially true for data from human participants which involve manipulations of level of consciousness through general anesthesia [26]. Ethics further limits recruitment of healthy participants for which there is no medical reason for inducing anesthesia or obtaining recordings. Due to these considerations, the use of independent, blinded data is particularly rare in consciousness research (but see [27]).

The issue of data availability in human anesthesia recordings can be circumvented by first applying our approach to simpler brains, such as fly brains. Recordings from flies can be obtained relatively cheaply with no clinical concerns, and, due to the relatively small brain, ($\sim 10^5$ neurons compared to $10^{11}$ for human brains; [28,29]), neural activity can be obtained simultaneously throughout the whole brain. Consequently, we can obtain high-quality recordings from many healthy flies. Using high-quality recordings from a relatively simple system also offers an advantage. That is, the identified time-series features can be more directly interpreted to understand underlying neural phenomena (compared to features identified from, e.g., recordings from the human scalp). Despite seemingly different neural architecture compared to mammals, flies seem to experience varying states of arousal, regulated in a similar way to mammals, such as sleep [30–33] and anesthesia [34,35]. Given these similarities and advantages described above, the fly serves as a useful model to begin to apply new data-driven approaches to discriminating consciousness levels from univariate neural time series (see also [12,36,37]).

In this registered report, we aim to evaluate a massive, comprehensive set of individual time-series features, coming from multiple research fields, as potential markers of level of consciousness. Which univariate time-series features accurately and reliably distinguish between conscious levels? And do they correspond to previously proposed univariate measures of conscious levels? Or are there some conceptually unexplored time-series features which perform better? If no features reliably distinguish conscious levels, this would highlight the need for bivariate or multivariate features. These would include features such as coherence, Granger causality, [35,38], transfer entropy [39], Lempel-Ziv complexity (which can be applied both at the individual channel level as well as across multiple channels) [40], perturbational complexity index [41], etc. Alternatively, new measures derived from theories, such as integrated information, may be necessary [37,42]. Indeed, many theories of consciousness rely on interactions among parts, and would predict univariate features to be uninformative of conscious level.

Here, we compare the most comprehensive available set of scientific features, made available in the *hctsa* toolkit [18], searching for features that may warrant further exploration in the future as potential markers of consciousness. First, we search for features which reliably distinguish wakefulness from isoflurane anesthesia in a discovery dataset ($N = 13$

discovery flies) and which generalize to a blinded, independent dataset ($N$ = 12 flies with graded levels of anesthesia; $N$ = 18 flies with single level of anesthesia; and $N$ = 19 flies during sleep). Second, we search for features for which the direction of the effect of anesthesia (i.e., yield consistently higher or lower values in anesthesia versus wakefulness) is consistent across datasets. These directionally consistent measures could be useful in assessing level of consciousness when a subject's baseline is known. For these purposes, we apply and compare the *hctsa* features systematically. Critically, we validate them on recordings obtained from an independent set of flies which were blinded to the analysis team at the time of submitting the Stage 1 manuscript for this registered analysis. At the time of submitting the Stage 1 manuscript, our pilot results (on 2 of the evaluation flies) indicated that the performances of many features which had statistically significant performance in classifying wakefulness and anesthesia in the discovery dataset would not generalize to a second, independent, evaluation dataset, highlighting the importance of evaluating measures on independent datasets. Despite this, across the datasets, many features maintained their direction of the effect of anesthesia across the flies.

## Summary table

| Research question | What univariate time-series features (from *hctsa*) can serve as markers of level of consciousness ACROSS individuals? | What univariate time-series features (from *hctsa*) can serve as markers of level of consciousness WITHIN individuals? |
|---|---|---|
| Hypotheses | 1 hypothesis for each *hctsa* feature at each channel:<br>• Feature X classifies wake/anesthesia above chance | 1 hypothesis for each *hctsa* feature at each channel:<br>• Direction of effect of anesthesia for feature X is more consistent than chance |
| Sampling plan | Use existing data:<br>• 13 discovery flies (Canton S wild-type) × 8 2.25s epochs each of wake/isoflurane (published previously);<br>• 2 pilot evaluation flies (isoCJ1; previously unpublished) × 112 2.25s epochs each of wake/isoflurane;<br>• 10 multi-dosage evaluation flies (isoCJ1; previously unpublished), epochs from wake/isoflurane at same concentration as pilot evaluation flies, plus epochs from all 12 flies (including previous 2 pilot evaluation flies) at a second isoflurane concentration (number of epochs undisclosed to data analysis team, but expecting same/similar to pilot evaluation flies) and during recovery after isoflurane;<br>• 18 single-dosage evaluation flies (Canton S wild-type; previously unpublished), epochs from wake/isoflurane/post-isoflurane (number of epochs undisclosed to data analysis team);<br>• 19 sleep evaluation flies (Canton S wild-type; unpublished), epochs from wake/sleep (number of epochs undisclosed to data analysis team) | |
| Statistical analyses | Classification analysis, using a nearest-median classifier **trained on the discovery flies**.<br>• Obtain classifier accuracy on discovery flies (leave-one-fly-out validation) and evaluation flies<br>• Obtain significance by comparing classifier performance to random classification distribution ($\alpha$ = 0.05)<br>• FDR correction at each channel ($q$ = 0.05) | Consistency of wakeful epochs being greater/less than anesthesia epochs at each fly, based on direction of anesthesia effect **in the discovery flies** (see Methods section "Within-fly effect direction consistency")<br>• Obtain significance by comparing consistency to random consistency distribution ($\alpha$ = 0.05)<br>• FDR correction at each channel ($q$ = 0.05) |
| Pre-specified outcomes | The performance of feature X in discriminating wakefulness/anesthesia shows significant generalization across individuals and the feature is worth future investigation as a marker of conscious level if:<br>• It performs significantly in the discovery flies AND<br>• It performs significantly in the evaluation flies | The within-individual effect of anesthesia for feature X shows significant generalization across individuals, and the feature is worth future investigation as a marker of conscious level if:<br>• Consistency of the direction of the effect of anesthesia is significantly above chance in the discovery flies AND<br>• Consistency is significantly above chance in the evaluation flies, for the same direction as the discovery flies |

## Methods

### Data and preprocessing

We use already-collected local field potentials (LFPs) from fruit fly brains during wakefulness and during isoflurane anesthesia. We use four independent datasets: (i) a *discovery dataset* for initially identifying features which perform well at discriminating wakefulness from anesthesia; and (ii) a blinded *evaluation dataset* for assessing the generalizability of these features to a separate dataset, which manipulated level of consciousness through (a) multi-dose anesthesia ($N$ = 12), (b)

 

single-dose anesthesia (*N* = 18), and (c) sleep (*N* = 19). Fig 1 illustrates our data analysis pipeline for the two sets of flies. As our discovery dataset, we use previously published data from 13 flies [12,35,37]. As our blinded evaluation dataset, we use data from an additional 49 flies collected by TJ and BvS which were provided to AL and NT for analysis only after in-principle acceptance of the Stage 1 manuscript (initials refer to authors of this registered report). At the time of submission of the Stage 1 manuscript, 2 of the evaluation flies from the multi-dose anesthesia set were provided and used for pilot analysis, with the remaining flies being withheld for final evaluation after analysis methods are fixed.

### Discovery flies

For this dataset, we provide details relevant to this registered report (for full details see [35]). Thirteen laboratory-reared female *Drosophila melanogaster* (Canton S wild type 3–7 days post-eclosion) were collected under cold anesthesia and glued dorsally to a tungsten rod. Linear silicon probes (Neuronexus Technologies) were inserted laterally into the fly's eye [43]. Each linear probe consisted of 16 electrodes separated with a site separation of 25 μm, and covered approximately half of the fly brain. Recordings were made with a sampling rate of 25 kHz using a Tucker-Davis Technologies multichannel data acquisition system and downsampled to 1,000 Hz.

Recordings for each fly were obtained from two blocks: one block with 0 vol% isoflurane at the fly body (wake condition), followed by a block with 0.6 vol% isoflurane (anesthesia condition). Isoflurane was delivered from an evaporator to the fly through a rubber hose. Each block followed a series of air puffs, and consisted of 18 s of rest, 248 s of visual stimuli, another 18 s of rest, and a second series of air puffs. Isoflurane was administered following the last air puff of the first block, and flies were left to adjust to the new concentration for 180 s before beginning the second block. Flies in the wake condition responded to air puffs by moving their legs and abdomen, but were inert during the anesthesia condition. Flies regained responsiveness after isoflurane was removed, ensuring that flies were alive during the anesthesia recordings [36]. We use the data obtained in the 18 s period of each block corresponding to the rest period preceding the visual stimuli.

We bipolar re-referenced the LFPs by subtracting adjacent electrodes to acquire 15 signals which we refer to as "channels". Channel 1 refers to the channel positioned furthest into the fly brain. Finally, we segmented the 18 s period into 2.25 s segments, giving 8 epochs per fly and condition.

### Pilot evaluation flies

On 14/06/2019, the data-analysis team (AM, AL, and NT) was provided with 56 segments of 20 second spontaneous activity recordings from the data collection team (TJ and BvS). The 56 segments were known to the data analysis team as coming from 2 flies and from varied levels of anesthesia. The analysis team was initially blinded to the labeling of the segments, such that the source condition and fly of each segment was unknown. Further, the analysis team was blinded as to the distribution of segments coming from each fly or anesthesia condition (e.g., whether the 56 segments had an equal number of wake and anesthetized segments, or an equal number of segments from each fly), and to the specific variant of fly and the context in which the data had originally been collected.

However, these labels and information were made available (in June 2019) after early analyses using 18 s segments (corresponding to the original length of the segments from the discovery flies, instead of 2.25 s segments). We later deemed the 18 s approach inappropriate as we would be generalizing across-fly classification performance to within-fly classification performance (applying classifiers trained on a single epoch each of wakefulness and anesthesia from each discovery fly to multiple epochs from an individual pilot evaluation fly; see Section "Classification of conscious level"), before finalizing the full methods and parameters. With the labels, it was revealed that the 56 segments were equally divided into 14 segments of wakefulness or anesthesia for each of the two flies. It was also revealed that the flies were of a w2202 background (also called isoCJ1), which has a similar isoflurane sensitivity profile to the Canton-S wild-type fly (CS; [44]).

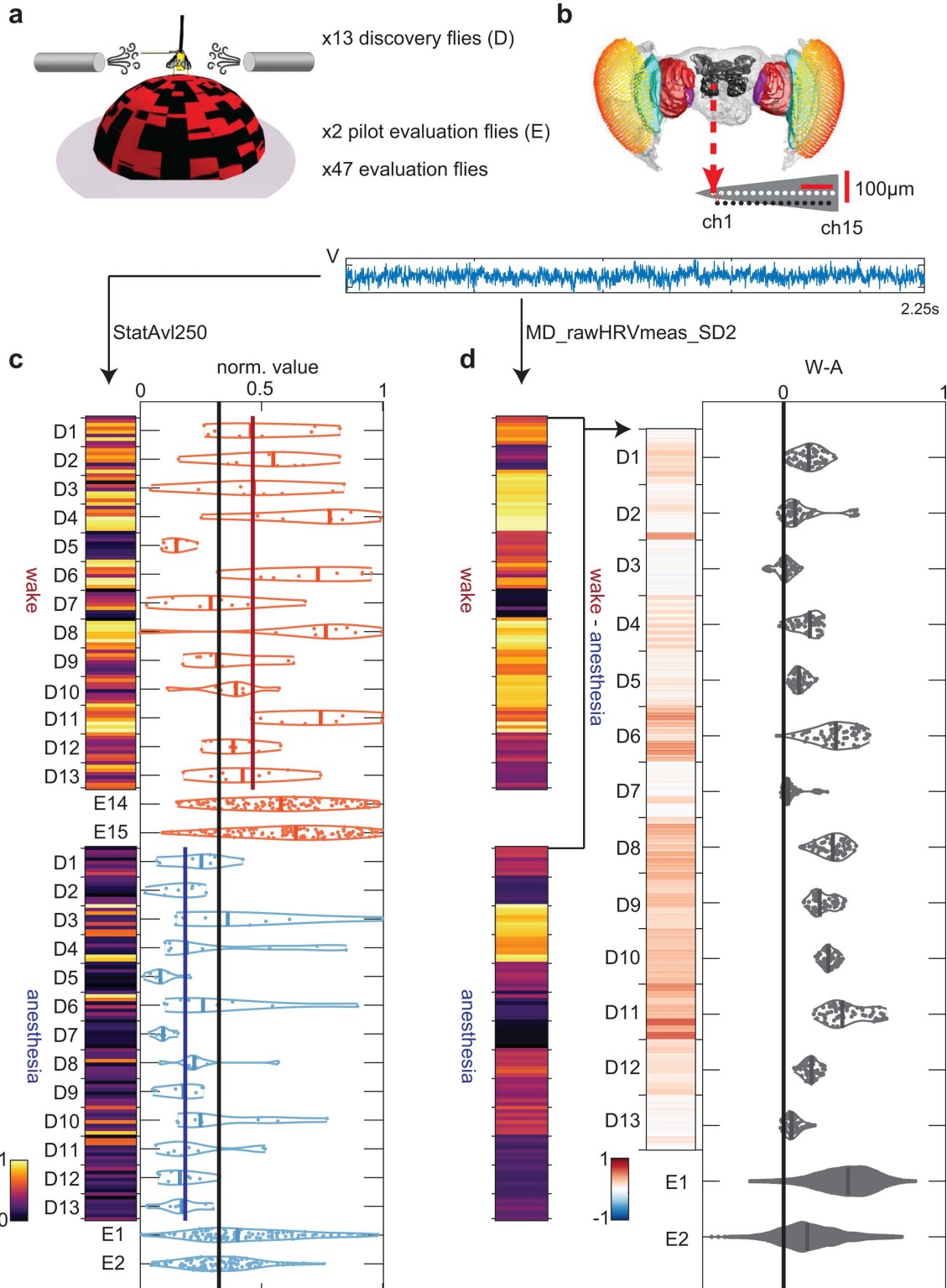

**Fig 1. Analysis pipeline for individual features in _hctsa_. ( a)** Flies were dorsally fixed to a rod and placed on an air-supported ball. Isoflurane was administered through a rubber hose. We use a discovery dataset of 13 flies to identify time-series features which discriminate wakefulness from anesthesia. We assess how the performances of these features generalize to an independent evaluation dataset consisting of 49 flies. We use 2 of these flies

to obtain pilot generalization performance for registering this analysis. **(b)** Local field potentials (LFPs) are obtained during wakefulness and anesthesia using linear multi-electrode arrays inserted laterally into the fly brain. **(c)** At a given channel and time-series feature (here we show the feature `Stat-Avl250`), we compute feature values for every epoch from each fly (each entry in the image plot corresponds to a scaled feature value from one epoch). We train a nearest-median classifier using the discovery (D) flies, where the threshold for classifying wakefulness (red) versus anesthesia (blue) is the middle point (black vertical line) between the median values of the two conditions (red and blue vertical lines). We assess the feature's across-fly classification performance on the discovery flies using a leave-one-fly-out cross-validation procedure. We assess the generalization of the feature's performance by classifying epochs from the evaluation (E) flies using its threshold as obtained from all the discovery flies. **(d)** As a weaker form of generalization, we also assess within-fly effect direction consistency by finding, for each wake epoch, the proportion of anesthesia epochs which have greater or lesser (depending on the direction of the effect of anesthesia for the feature as illustrated in **(c)** feature values. We visualize this here for a second feature by showing the within-fly differences between scaled feature values. Each entry in the rightmost image plot gives the difference between every combination of one wake epoch and one anesthesia epoch from the same fly. Images in **(a)** and **(b)** obtained from [36]. Source data available on OSF (https://osf.io/8wvsq/?view_only=8a056d1c573b4f23a6cf6cea8b976ddb).

The following technical details of the recordings were available to the data analysis team, to enable equal pre-processing of signals. Electrophysiological data were recorded at 25 kHZ, down sampled to 1,000 Hz. Next, LFPs were bipolar re-referencing by subtracting adjacent unipolar channels ($n = 16$) to acquire 15 channels.

The exact details originally provided to the analysis team are available at https://osf.io/bq5ry/?view_only=3789097395c1419db2a9eb615bc1effe.

## Final evaluation flies

Final evaluation data were provided to the data analysis team after in-principle acceptance of the Stage 1 manuscript. We describe the information disclosed by the data collection team (TJ and BvS) to the data analysis team at the time of writing Stage 1 of this registered analysis, followed by additional details which were provided after in-principle acceptance. The evaluation data consists of three datasets as follows. The electrode preparation for each dataset followed the same protocol as described in [45]. At time of submission of the Stage 1 manuscript, the analysis strategy was fixed, and final evaluation data had not been provided to the analysis team. The teams agreed that disclosing the following information would not affect the outcome of the results.

## Multi-dosage evaluation flies

The multi-dosage evaluation dataset consists of 12 female isoCJ1 flies, which were administered isoflurane at two concentrations. The two pilot evaluation flies described previously and analyzed in Stage 1 of this registered analysis were taken from this dataset (using epochs from wakefulness and one concentration of isoflurane anesthesia). Epochs from this dataset were obtained after an air puff stimulus and consist of five possible conditions: (1) wakefulness; (2) isoflurane concentration A; (3) isoflurane concentration B; (4) post-isoflurane (after isoflurane administration, but before flies are fully awake); and (5) recovery (when flies are fully awake and responsive after isoflurane). At the time of submission of the Stage 1 manuscript, the isoflurane concentrations A and B were not disclosed to the data analysis team.

After in-principle acceptance of the Stage 1 manuscript, the recordings from these flies were provided, and the following details were revealed. Recording setup and anesthetic delivery was similar to the discovery flies, with 16-electrode linear silicon probes with site separation of 25 μm inserted laterally into the fly's eye, and recordings being made at 25 kHz and downsampled to 1,000 Hz. Probes were inserted such that the outermost electrode was within the fly's eye (this was also the procedure for the flies in the single-dosage and sleep evaluation flies), unlike in the discovery flies where the outermost electrode was positioned just outside the eye. For each fly, 14 chunks of 20 s recordings, bipolar re-referenced in the same manner as the discovery flies, were provided for each condition.

Similar to the discovery flies, recordings were obtained in sequential blocks corresponding to the conditions outlined previously. Each block consisted of a 5 min period of darkness, followed by an air puff and 15 min of red ambient

lighting and visual flickering stimuli. Blocks were separated by 1 min, with an air puff 30 s after the end of each block. The provided recordings corresponded to the 5 min period of darkness at the beginning of each block. We divided each 20 s chunk into eight 2.25 s epochs, discarding the last 2 s of each chunk, to match the epoch duration of the discovery flies. It was revealed that the isoflurane concentrations A and B at the fly body were 0.6 and 1.2 vol%, respectively, as estimated through gas chromatography. However, due to the increased concentration of isoflurane and longer duration of exposure, an increased vacuum flow compared to the discovery flies (16 L/min, compared to 9 L/min in the discovery and single-dosage evaluation flies) was used to clear the room of anesthesia. As such, actual anesthetic concentration is likely to have been lower. Unlike the discovery flies, recordings were obtained in the absence of ambient lighting.

Upon conducting our analyses, we found that the distribution of feature values varied greatly between the first eight and last four flies. Upon further investigation, the analysis team discovered that the flies had been recorded in two batches. Originally, eight flies were recorded. An additional four flies were subsequently recorded from after a delay of a year. While to our knowledge the recording and experimental setups were unchanged, the distribution of feature values varied greatly between the two subsets of flies. From inspecting power spectra, we suspected that a low pass filter may have been inadvertently in use during the recording of the last four flies (Fig B in S1 Text). Thus, we separate out these two subsets of flies. We refer to the first eight flies as MD8 and the last four flies as MD4.

As electrode probes were inserted deeper into the evaluation flies than for the discovery flies, we considered offsetting the channels of the evaluation flies (i.e., all the flies from the multi-dosage, single-dosage, and sleep evaluation flies) to better align them with the channel locations of the discovery flies. However, we did not find evidence to suggest that such an offset would better align the channels (S1 Text).

## Single-dosage evaluation flies

The single-dosage evaluation dataset consists of 18 female Canton S wild-type flies, which were administered isoflurane at a single concentration. Epochs from this dataset were obtained after an air puff stimulus and consist of four possible conditions: (1) wakefulness; (2) isoflurane concentration C (which may or may not be equal to isoflurane concentrations A and B in the multi-dosage evaluation flies); (3) post-isoflurane; and (4) recovery.

After in-principle acceptance of the Stage 1 manuscript, the recordings from these flies were provided, and the following details were revealed. Flies were 3–10 days post-eclosion. Fly preparation and anesthesia delivery was again similar to the discovery flies, with 16-electrode linear silicon probes with site separation of 25 μm inserted laterally into the fly's eye, and recordings being made at 25 kHz and downsampled to 1,000 Hz. For each fly, 18 s from each condition was provided.

Again, similar to the discovery flies, recordings were obtained in sequential blocks corresponding to each condition. Each block consisted of an airpuff, followed 30 s later by visual stimuli lasting 18 min and 18 s, followed by another 30 s of rest. Isoflurane was then administered and flies were left for 180 s to adjust to the new concentration before starting the next block. The provided 18 s recordings correspond to 18 s preceding the visual stimuli. We bipolar re-referenced the LFPs in the same manner as the discovery flies and segmented the 18 s recordings into 2.25 s epochs to match the epoch duration of the discovery flies.

## Sleep evaluation flies

The sleep evaluation dataset consists of 19 female Canton S wild-type flies. Recordings were obtained during wakefulness and periods of sleep. Hence, epochs from this dataset consist of two possible conditions: (1) wakefulness, and (2) sleep. While the body of the flies were tethered, their limbs were free to move. We obtained video recordings of the flies to detect and quantify the movement of the flies [36]. With this movement detection, we defined sleep using the commonly accepted criterion of periods of immobility lasting 5 min or longer [30].

After in-principle acceptance of the Stage 1 manuscript, the recordings from these flies were provided, and the following details were revealed. Flies were 1–2 days post-eclosion to maximize survivability across 12 h. Recordings for different flies started at different times spread throughout the 24-h day to capture bouts of sleep during both the day and night. As for the previous datasets, fly preparation was similar to the discovery flies. Linear 16-electrode silicon probes with site separation of 25 μm were inserted laterally into the fly's eye, and recordings were made at a sampling rate of 25 kHz and downsampled to 1,000 Hz.

Along with the electrophysiological recording, video recordings of the flies in profile view were captured to identify sleep bouts. Recordings were made using a Scopetek DCM130E with a Navitar zoom lens (coupler 1–6,010, adapter tube 1–6,020) under infrared light. The resulting videos were analyzed using OpenCV to identify periods of continuous periods of immobility, using the procedure described in [46]. For each fly, one recording block containing at least 5 min of continuous immobility was provided for analysis.

With one block of sleep per fly, we extracted a period of 18 s (wake) immediately before the onset of the sleep bout, and 18 s ending 2 min after the onset of the sleep bout (sleep). Fig C in S1 Text illustrates the extracted temporal locations relative to the recording block provided for each fly. We selected these temporal locations due to varied sleep and recording lengths for each fly in the provided recordings. In the same manner as for the other flies, we bipolar re-referenced the LFPs and segmented the 18 s period into 2.25 s segments.

## Local field potential pre-processing

The data analysis team subtracted the mean voltage from each epoch of the discovery and pilot evaluation flies, and then removed line noise from each epoch using the **rmlinesc.m** function of the Chronux toolbox (http://chronux.org/; [47]) with 9 tapers, a time-bandwidth product of 5, and zero-padding factor 2. As a sanity check, we performed visual inspection of power spectrum plots after pre-processing to confirm the removal of line noise. These same pre-processing steps were also applied to the final evaluation flies.

## Feature-based time-series analysis using *hctsa*

We extracted 7,702 time-series features from each epoch and bipolar re-referenced channel of the discovery and pilot evaluation flies using *hctsa* (v1.03; [18]) on MATLAB 2017b. For a given time series, *hctsa* extracts a vast set of 7,702 univariate time-series features from analysis methods developed in a wide range of scientific disciplines, including nonlinear physics, biomedicine, economics, and neuroscience. *hctsa* broadly groups these features into several general themes, such as distribution, correlation, information theory, stationarity, and so on. Within these themes, features are further grouped into "master operations", which implement computations which are relevant to groups of time-series features. In this way, individual features are obtained by running master operations while specifying a range of parameters. An example of this are the features SP_Summaries_welch_wmax_<P>, which compute spectral edge frequency at P, with P=5%, 10%, 25%, and 95%.

Not all of the available time-series features could be extracted successfully from our datasets. For example, the class of features derived from the *hctsa* function DN_CompareKSFit includes fits of the data to a beta distribution, which assumes values between 0 and 1, an assumption that is not fulfilled by our data and consequently returns missing (NaN) values. To filter out these cases, we excluded any feature which returned NaN across all time series for a given channel in the discovery flies. This reduced the set of features down to an average of 6,860 features across the 15 channels (ranging from 6,657 to 7,004). We further excluded features which returned a constant value across all time series for a given channel in the discovery flies because they are uninformative for classification, reducing the set of features again to on average 6,764 features across the 15 channels (ranging from 6,560 to 6,908).

While we analyze raw *hctsa* features, the range of values varies greatly across features, and some features include infinity values (which we keep, as they can be used in our classification analysis, see Section: "Classification of conscious level"). Where specified, we visualize scaled feature values using an outlier-robust sigmoidal transformation, which maps

values of all epochs for a given feature to the unit interval [48]. We scale feature values in the evaluation flies based on the scaling parameters from the discovery flies.

## Classification of conscious level

We use single-feature classification analysis at each channel to compare the performance of each individual time-series feature in distinguishing wakefulness from deep anesthesia (i.e., highest dosage in the multi-dose data) and sleep. If a feature distinguishes conscious level, it should have high classification performance in the discovery flies which generalizes to the evaluation flies. To account for features which can return infinity values, we employ nearest-centroid classifiers, with class medians as centers.

We first trained and cross-validated each feature's classifier on the discovery flies. For a given channel and feature, we employed a leave-one-fly-out cross-validation procedure on the evaluation flies. Specifically, at each cross-validation iteration, we trained a classifier on all 8 epochs of wake and anesthesia from 12 flies, and tested on 8 epochs of wake and anesthesia on the remaining fly. Each classifier consists of: (1) a threshold, the middle point between the median feature value for wakefulness and anesthesia as obtained from the training set; and (2) a direction indicating whether points above the threshold should be classified as wakeful or anesthetized (and vice versa).

After obtaining cross-validation accuracies on the discovery flies, we finally obtained classifiers for each feature and channel by training on all epochs from all flies in the discovery dataset. We validate and report the performance of these classifiers on the pilot evaluation flies ($N = 2$, in the Stage 1 manuscript) and final evaluation datasets (total $N = 49$ evaluation flies, after in-principle acceptance).

In both experimental and applied settings, feature values can drift across datasets, due to uncontrollable changes in experimental settings or unknown sources of group differences. Some features may be sensitive to such uninteresting drift, which can be corrected automatically. To account for such drift, we report classifier performances both before and after normalizing feature values in the evaluation flies to match the mean and standard deviation of values in the discovery flies. We normalized each pilot evaluation fly by transforming feature values into $z$-scores, ignoring infinity values (which remained as infinity values after normalization), using the mean and standard deviation across all epochs from the other pilot evaluation fly. We then back-transformed the resulting $z$-scores using the mean and standard deviation across all epochs from all the discovery flies. We normalized the final evaluation datasets in a similar manner, by transforming feature values into $z$-scores using the mean and standard deviation across all epochs from each set of the evaluation flies and back-transforming using the mean and standard deviation across all epochs from the discovery flies.

We determined if a feature's classifier discriminates wake and anesthesia significantly better than chance by comparing each feature to a random classification distribution at the $a = 0.05$ level. We corrected for multiple comparisons at each channel using the false discovery rate (FDR) correction [49] to account for potential positive dependency among the tests, which is likely to be the case as there are features in *hctsa* which are expected to give similar results (such as `CO_Auto-Corr` features which compute autocorrelation at different lags) [50,51]. We obtained random-classification distributions for the discovery and pilot evaluation flies by repeatedly classifying discovery or evaluation epochs randomly, with equal probability (as there are 7,702 potentially available features in *hctsa*, we repeated this random classification $N = 7,702$ times to estimate the null distribution). We expected that features which reflect some process underlying change in conscious level will have significant classification performance which persists through cross-validation on the discovery flies to the final evaluation flies.

Upon receiving the datasets for Stage 2 analysis, we made additional methodological decisions as follows. As the evaluation datasets consisted of multiple wake and/or multiple unconscious conditions, we evaluated classification performance by, for a given dataset, pairing each wake state with each unconscious state. For the multi-dosage and single-dosage evaluation flies, we considered the isoflurane and sleep conditions as unconscious, and the others as wake. This gave a total of 6, 2, and 1 possible condition pairs for multi-dosage, single-dosage, and sleep flies, respectively

(totaling 16 condition pairs, after splitting the multi-dosage flies into two groups and including the wake-anesthesia pair in the discovery flies). To obtain the performance of a given feature at a given condition pair, we used all the predictions from the wake condition and the unconscious condition. The number of epochs among all conditions for a given dataset was always equal, hence chance performance was always 50% for all condition pairs. As we found no features to generalize across all the condition pairs (i.e., achieve significant classification performance or consistency), we proceeded to limit our analyses to the condition pairs which we deemed as the most similar to the discovery flies. Given the experimental details provided for each dataset, we considered the pre-anesthesia and 1.2 vol% isoflurane conditions in the multi-dosage evaluation flies to be most similar to the wake and anesthesia conditions in the discovery flies. For the single-dosage evaluation flies, we considered the pre-anesthesia and 0.6 vol% isoflurane conditions to be the most similar to the discovery fly conditions.

## Within-fly effect direction consistency

In assessing generalizability, it is possible that the effect of anesthesia (relative to wake) is highly consistent within individuals, even when features do not classify well across subjects. This is relevant in scenarios where, such as in this registered report, there may be variability in the exact placement of electrodes among individuals, an effect which cannot be corrected even with our group-level batch normalization. Values may further vary among individuals due to factors such as exact experimental setups and baseline arousal states. To address this, we assessed a weaker form of generalization—whether a feature is predictive of the relative difference between conscious levels within an individual fly—and report the consistency of the direction of the effect of anesthesia (after receiving the correct wake/anesthesia/sleep labels in the case of the final evaluation flies).

Specifically, at a given feature, fly, and channel, we obtained for each wakeful datapoint the proportion of anesthetized data points which lie below it. Because the direction of the effect of anesthesia is not necessarily the same across features and channels, we first assigned directionality labels based on the median wakeful and anesthesia values in the 13 discovery flies (i.e., based on the direction component of the classifiers described above). For a given feature and channel, we gave a label of 1 if the median wakeful value was greater than for anesthesia, and −1 otherwise. We then multiplied feature values by these labels, flipping the direction of the effect of anesthesia when the median wakeful value is lesser than the median anesthesia value and making the analysis uniform across features and channels. Finally, we report the average proportion across all wakeful epochs and flies.

In a similar way as for testing for significance of classification performance, we used permutation testing to determine if the within-fly effect direction consistency of a feature was significantly better than chance. We obtained reference chance distributions for the discovery flies and pilot evaluation flies by repeatedly ($N = 7,702$) randomly assigning the portion of anesthesia epochs which are below each wakeful epoch, with equal probability, and averaging across wakeful epochs and flies. We compare each feature to the distribution at the $\alpha = 0.05$ level, correcting for multiple comparisons at each channel using FDR correction. We again limited our analysis to the conditions in the evaluation flies which were most similar to the discovery flies, for the same reasons as previously described for the classification analysis.

## Pilot results

We investigate if any of the time-series features in *hctsa* individually serve as a potential measure of level of conscious arousal in independently obtained recordings from fly brains. We first assessed the performance of *hctsa* features which we applied to a discovery dataset of previously published fly brain recordings ($N = 13$) [12,35–37]. Then, to assess generalizability, we apply classifiers trained on the discovery flies to recordings obtained from an independent set of pilot evaluation flies ($N = 2$). Upon in-principle acceptance of this registered analysis, we repeated the analyses conducted on the pilot evaluation flies on a final set of evaluation flies ($N = 47$), reporting the features which consistently perform well in distinguishing wakefulness from anesthesia and sleep at all recording locations across all the flies.

## Classification of conscious level

We first extracted 7,702 time-series features from the initial discovery flies using *hctsa*, yielding 6,560–6,908 valid features across the 15 channels (*M* = 6,764). Fig 2A shows a matrix of feature values extracted from Channel 6 in the discovery dataset. We first visually inspected this feature matrix to inspect trends across features and flies. To facilitate interpretation, we first sorted the order of the features according to hierarchical clustering using correlation distances between features, across time series. This revealed two clear clusters of features, one with values which are generally greater during wakefulness (columns roughly 500–1,500), and one with values which are generally greater during anesthesia (columns roughly 4,500–5,500). Features in each of these clusters would likely achieve similar classification accuracies.

Having reordered the features, groupings across rows corresponding to epochs from individual flies became apparent. This indicated strong within-fly correlations of feature values but weak correlations across flies, suggesting that few features, if any, would generalize across all the flies. Overall, our visual inspection of the similarities across features and similarities across flies suggested that many features could individually achieve better-than-chance classification accuracy. However, there appeared to be no clear cluster of features which would perfectly discriminate wakefulness from anesthesia in all of the flies.

While Fig 2A set up our global expectations visually, there may have been features outside the visually clear clusters which also distinguish wakefulness from anesthesia extremely well. To reveal such features, we next quantified the across-fly classification performance (within the discovery flies). For each feature, we classified wakeful from anesthetized epochs using a nearest-median classification rule. We assessed the statistical significance of the cross-validation accuracy of each feature by comparing it to a distribution of accuracies resulting from random classification (see Methods). For Channel 6, this yielded 3,089 features which performed significantly better than chance (*p* < .018). The best-performing feature, an index of mean stationarity (*hctsa* feature: `StatAvl250`; [52]), achieved a mean classification accuracy of 76% (*SD* = 12% across 13 cross-validations; Fig 2B). Upon performing the classification analysis for each of the remaining 14 channels, we found features to perform heterogeneously across the channels. Overall, the average classification accuracy achieved across channels tended to be much lower than that achieved by individual channels. For example, the across-channel average of the mean cross-validated accuracy of `StatAvl250` was 63% (*SD* = 6% across 15 channels).

Indeed, the number of significant features varied greatly across the channels, ranging from 14 to 2,948, with channels closer to the periphery tending to have fewer significantly performing features (Fig 2B). We found the greatest number of significant features, along with the most accurately classifying features, to occur at Channels 5 and 6, corresponding roughly to the protocerebrum. This is consistent with our previous analyses on this dataset, which reported better discrimination between wakefulness and anesthesia in some but not all channels [12,35].

We next sought to determine how well the performance of features would generalize to an independent evaluation set of flies. While the overall recording procedure was known by the data analysis team to be similar to that of the discovery flies, the exact experimental methods were not revealed at the time of submitting this registered analysis (see Methods). We finalized the training of classifiers by obtaining thresholds based on all 13 discovery flies. As a pilot for this registered analysis, we applied these classifiers to recordings from 2 flies (out of a total of 12 evaluation flies). Across the 15 channels, we found an additional 48–416 (*M* = 180) features to either output a `NaN` or have a constant value across epochs in the pilot evaluation flies.

Fig 2C shows how classification performance at Channel 6 in the discovery flies generalizes into the pilot evaluation flies. The best performing feature at Channel 6 in the discovery flies, `StatAvl250`, attained a much-reduced accuracy of 63% (green circle). Meanwhile, several features attained higher performance than in the discovery flies. The feature with the best performance at Channel 6 in the pilot evaluation flies, which quantifies the relative low-frequency power in the Fourier power spectrum (*hctsa* feature: `SP_Summaries_welch_rect_logarea_2_1`), attained 76% accuracy, despite attaining 62% (*SD* = 14% across cross-validations) in the discovery flies (red circle). The best-performing features across

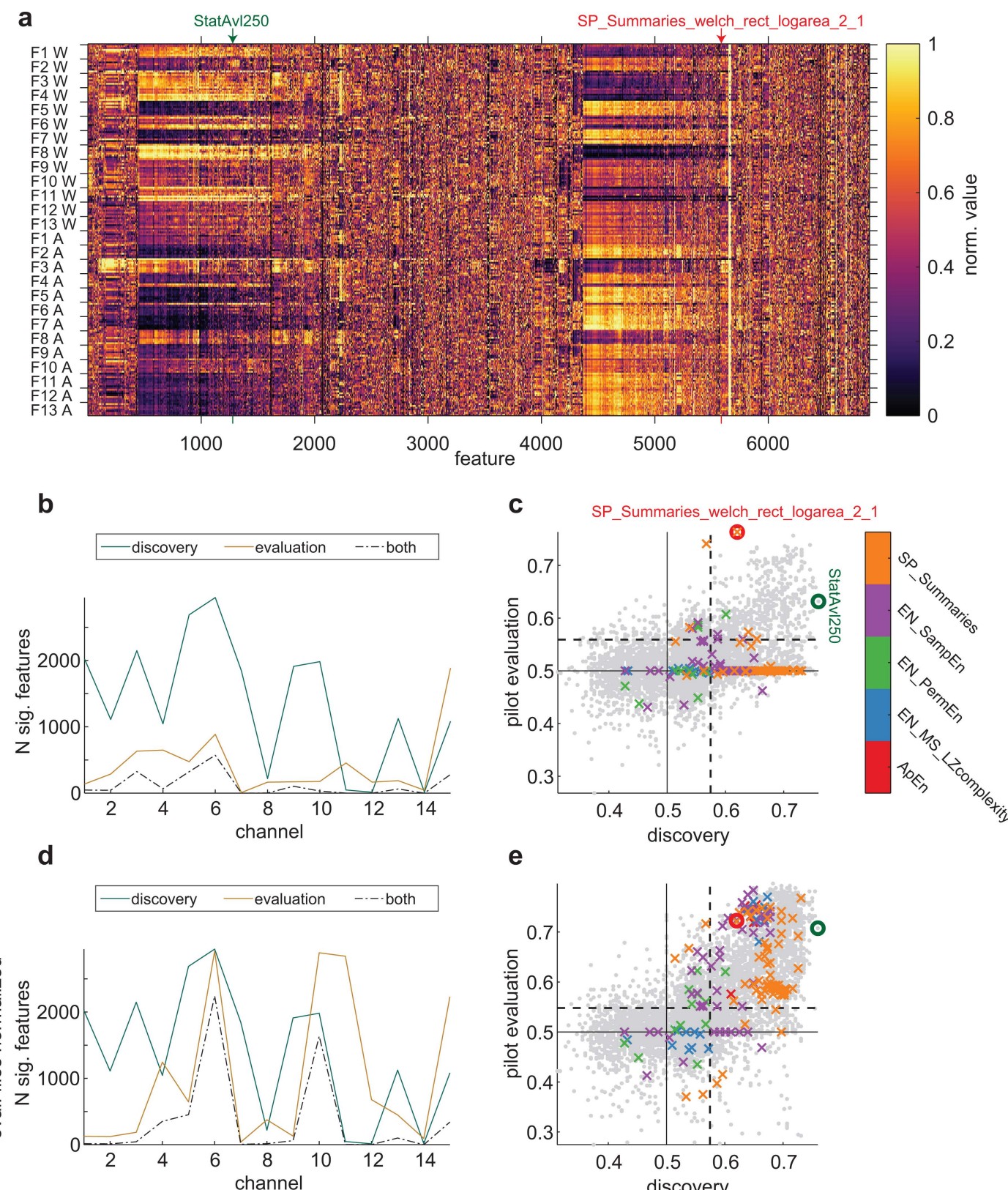

**Fig 2. Classification performance of *hctsa* features. ( a)** Values of *hctsa* features in the discovery flies, at Channel 6. Each row corresponds to an individual 2.25 s epoch, from 13 flies (F) during wakefulness (W) and anesthesia (A). Each row displays scaled values for all valid features for the

channel. Features (columns) are ordered based on hierarchical clustering using correlation (across time series) distance between normalized features. This ordering places features with highly correlated values across the dataset close to each other. Arrows indicate the features which attained the highest classification performance in the discovery (green) and pilot evaluation (red) flies. **(b)** Number of features which achieved statistically significant classification performance at each channel, in the discovery flies (blue line), pilot evaluation flies (orange line), and in both the discovery flies and pilot evaluation flies (broken black line). **(c)** Correlation of classification performances between the discovery (x-axis) and pilot evaluation flies (y-axis). Each dot represents the classification performance of one of the 6,800 features shown in **a**). Solid horizontal and vertical lines indicate chance classification performance (= 0.5). Dashed horizontal and vertical lines indicate the thresholds for statistically significant across-fly classification performance in each set of flies (see Methods). Dots located in the top right quadrant are the features which successfully classified wake from anesthesia across both the discovery and pilot evaluation flies. Circled are the features which attained the highest performance in the discovery (green) and pilot evaluation (red) flies (corresponding to the features pointed to in **a**). Coloured x's indicate performance of features related to previously described measures of conscious level (features related to spectral power across frequency bands and spectral edge frequency, `SP_Summaries`; sample entropy, `EN_SampEn`; permutation entropy, `EN_PermEn`; Lempel-Ziv complexity, `EN_MS_LZcomplexity`; approximate entropy; `ApEn`, as indicated by the color bar). **(d)** Number of features which achieved statistically significant classification performance at each channel, as in **b**), but after normalizing the pilot evaluation flies. **(e)** Correlation of classification performances between the discovery and pilot evaluation flies, as in **c**), but after normalizing the pilot evaluation flies.

all the channels in the pilot evaluation flies were related to signal variance at Channel 5 (including root-mean-square, *hctsa* feature `rms`, and standard deviation, `standard_deviation`), and also had greater performance than in the discovery flies, attaining 91% accuracy, compared to 71% ($SD = 21\%$ across 13 cross-validations, for both features).

For comparison, Fig 2C also shows the performance of *hctsa* features which are related to previously described indicators of conscious level in human electroencephalographic (EEG) recordings. These include features related to spectral power across frequency bands, spectral edge frequencies, and spectral entropy (`SP_Summaries`), sample entropy (`EN_SampEn`), permutation entropy (`EN_PermEn`), single-channel Lempel-Ziv complexity (`EN_MS_LZcomplexity`), and approximate entropy (`ApEn`) [13,14,53]. Surprisingly, the vast majority of these features did not classify wakefulness from anesthesia significantly for both the discovery and pilot evaluation flies. While we leave the interpretation of high-performing features until after the final analysis, it is notable that most of these features are related to the variance of the voltage fluctuations, which is consistent with previous literature on the effects of anesthesia on fly LFPs [35].

Generally, however, across the channels, we found a drastic drop in the number of features with statistically significant classification accuracy (Fig 2B). This suggested that features overall performed worse in the evaluation flies, and that their performance was again heterogeneous across channels. Across the channels, the number of significantly performing features was substantially less in pilot evaluation flies, ranging now from 9 to 867. Further, after restricting to the set of features which yielded significant cross-validation accuracies in the discovery flies, the number of significant features dropped even further, ranging across the channels from 0 to 561. Fig 2C illustrates this drop also for the majority of previously described indicators of conscious level in EEG, particularly for spectral features which performed well in the discovery flies but at chance level in the pilot evaluation flies. This result alerts us to the danger of interpreting cross-validation accuracies of the discovery flies as an estimate of the true generalization accuracies, which can only be evaluated using an independent dataset. We will discuss the implication of this finding in Discussion in the Stage 2 manuscript. Upon final data analysis, we provide the classification performance of each significant feature, at each channel in https://osf.io/8wvsq/?view_only=8a056d1c573b4f23a6cf6cea8b976ddb.

Given that the performance of so few features generalized to the pilot evaluation flies, we next repeated the above analysis, but after normalizing feature values in the pilot evaluation flies to match the distribution of values in the discovery flies (see Methods). This normalization ensures that the ranges of feature values in the pilot evaluation flies match those in the discovery flies. With normalization, many more features performed significantly in the pilot evaluation flies, ranging from 38 to 2,892 features across the 15 channels (which reduced to 0–2,223 features after restricting to features which also performed significantly in the discovery flies; Fig 2B), with many features which previously performed at chance level in the pilot evaluation flies performing significantly above chance after normalization (Fig 2D). The highest performing feature in the discovery flies, `StatAvl250`, achieved an improved accuracy of 71%. With normalization, the

best performing feature across all the channels in the pilot evaluation flies was also a statistical moment of the signal, this time kurtosis (feature `DN_Moments_raw_4` at Channel 5, achieving 92% accuracy). Upon final data analysis, we also provide the classification performance of each significant feature after normalization, at each channel in https://osf.io/8wvsq/?view_only=8a056d1c573b4f23a6cf6cea8b976ddb.

### Within-fly effect direction consistency

Given that the across-flies classification performance of many features in the discovery flies did not generalize to the pilot evaluation flies, we next assessed a weaker form of generalization. Even though features may not classify well across subjects, features for which the effect of anesthesia is highly consistent within individuals may still be useful for clinical assessment of conscious level. This is especially true for individual subjects whose baseline neural activity is available (e.g., before anesthetic induction). So, for each feature, we assessed the within-fly effect direction consistency of anesthesia. For an individual fly, consistency would be 1 if the direction of change of feature values from wake to anesthesia (i.e., either an increase or decrease) is conserved for every pairing of wake and anesthesia epochs (see Methods). In other words, for a given feature in a given channel, if the value for wake minus anesthesia is always above 0 for any pair of one wake and one anesthesia epoch (or vice versa), then we consider such a feature as a perfect measure of consciousness when a baseline measurement is available.

Fig 3A illustrates the within-fly effect direction consistency for each feature for the discovery flies, again at Channel 6, by showing the differences in feature values between wake and anesthesia epochs. Overall, the direction of the effect of anesthesia for each feature appeared to be reliable within individual flies, as we expected. However, strikingly, the direction of the effect of anesthesia seemed to be consistent even across flies, despite mediocre classification performance (e.g., due to differing baseline values at each fly). Visually, there appeared to be two clusters of features (from column 500–1,500 and from 4,500 to 5,000) with high consistency.

We assessed the statistical significance of each feature, this time by comparing its consistency to a distribution of consistencies for randomly labeled epochs (see Methods). For Channel 6, this gave 3,882 features which were more consistent than chance ($p < .027$). The feature with the highest consistency at Channel 6, as well as on average across all the channels (a measure of variability along the identity line when plotting, in our data, time samples against their immediate future values, *hctsa* feature: `MD_rawHRVmeas_SD2`; [54]) had a consistency of 0.94 (i.e., on average, each wakeful epoch from an individual fly had a greater value than 94% of anesthesia epochs from the same fly), which previously attained an across-flies cross-validation accuracy of 68% ($SD = 20\%$). Across the 15 channels, in general we found many more features to have significant within-fly consistency (2,474–3,882; Fig 3B), compared to across-flies classification.

We next assessed how the within-fly effect direction consistencies generalized to the pilot evaluation flies. We computed consistencies in the pilot evaluation flies, taking into account the direction of the effect of anesthesia observed in the discovery flies. Hence, if wakeful and anesthesia epochs were perfectly separable in the same direction as the discovery flies, consistency would be 1. However, if they were perfectly separable in the opposite direction to the discovery flies, consistency in pilot evaluation flies would be 0.

Fig 3C shows how within-fly effect direction consistency at Channel 6 in the discovery flies generalizes to the pilot evaluation flies. Unlike for across-flies classification, within-fly consistencies between the discovery and pilot evaluation flies seemed to be strongly positively correlated, including features related to previously explored measures of conscious level. This indicates that the within-fly consistency of many more features generalized to the pilot evaluation flies. The feature with the highest consistency in the evaluation flies, root-mean-square (*hctsa* feature: `rms`), also achieved high consistency in the pilot evaluation flies (0.91, red circle and arrow). Across the 15 channels, the number of significantly consistent features seemed to vary more in the pilot evaluation flies, ranging from 463 to 3,899 (Fig 3B). This range reduced to 176–3,049 after restricting to the set of features which also had significant consistency in the discovery flies.

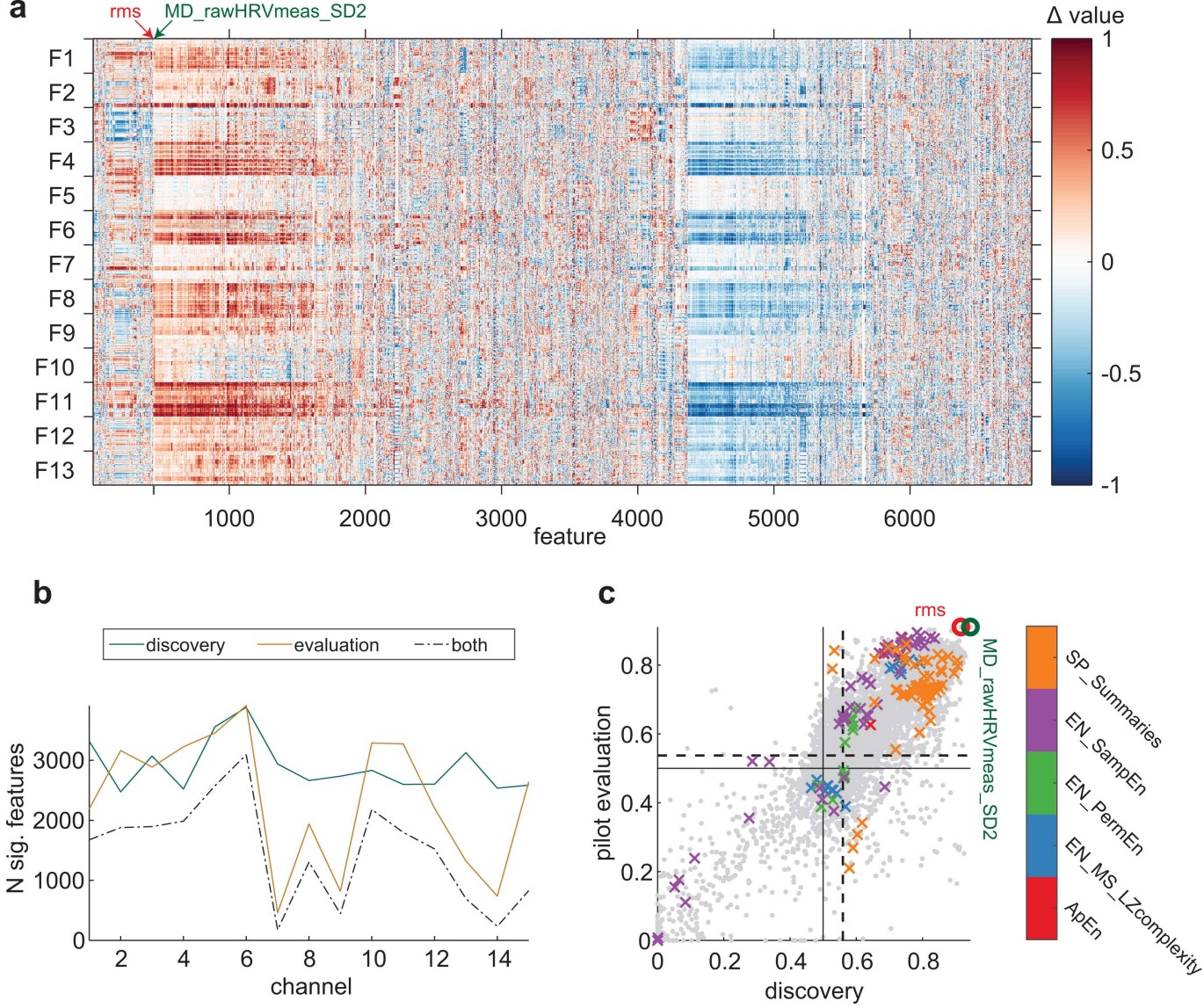

**Fig 3. Within-fly effect direction (wake – anesthesia) consistency of *hctsa* features. ( a)** Differences in scaled *hctsa* values between wakefulness and anesthesia in the 13 discovery flies (F), at Channel 6. Each row displays the difference between a wakeful and anesthesia epoch from the same fly, for all valid features for the channel. Features (columns) have the same ordering as in Fig 2A. Arrows indicate the features which attained the highest classification performance in the discovery (green) and pilot evaluation (red) flies. **(b)** Number of features which achieved statistically significant consistency at each channel, in the discovery flies, pilot evaluation flies, and in both the discovery flies and pilot evaluation flies. **(c)** Correlation of within-fly effect direction consistencies between the discovery (x-axis) and pilot evaluation flies (y-axis). Each dot represents the consistency of one of the 6,800 features shown in **a**). Solid horizontal and vertical lines indicate chance consistency (= 0.5). Dashed horizontal and vertical lines indicate the thresholds for statistically significant consistency in each set of flies (see Methods). Dots located in the top right quadrant are the features which were significantly consistent across both the discovery and pilot evaluation flies. Circled are the features which attained the highest consistency in the discovery (green) and pilot evaluation (red) flies. Colored x's indicate performances of features related to previously described measures of conscious level, as described in Fig 2C and indicated by the color bar.

Notably, the decrease in the number of significant features for within-fly consistency in the pilot evaluation flies was less pronounced than for across-fly classification performance. Like across-fly classification, there were more significant features for within-fly consistency at the central than peripheral channels. Overall, these results indicate that many features

could be informative and consistent in terms of changes within a single fly due to anesthesia without being strong, absolute measures of conscious level across flies. We will revisit the implication of this finding in Discussion after the final data analysis. Upon final data analysis, we provide the consistencies of each significant feature, at each channel in https://osf.io/8wvsq/?view_only=8a056d1c573b4f23a6cf6cea8b976ddb.

## Results

As described in the Stage 1 Manuscript, we repeated the analysis as described in Methods and Pilot results, extending and updating the analyses done on the pilot evaluation flies to the full set of 47 evaluation flies. We investigated each of the time-series features in *hctsa* as potential measures of level of conscious arousal. The number of time-series features extracted using *hctsa* which we considered to be valid varied across the datasets. The multi-dosage evaluation flies yielded 7,401–7,420 valid features across the 15 channels ($M = 7,413$), the single-dosage evaluation flies yielded 6,570–6,923 valid features ($M = 6,632$), and the sleep evaluation flies yielded 6,492–6,963 valid features ($M = 6,629$). Considering only features which were valid across all the evaluation flies and the discovery flies yielded 6,367–6,742 ($M = 6,480$) valid features altogether across the 15 channels. To evaluate the generalizability of features identified in the Stage 1 Manuscript, we applied the classifiers previously trained on the discovery flies during the Stage 1 analysis to recordings from the independent set of evaluation flies ($N = 49$) which were previously inaccessible to the data analysis team (AL, BF, and NT). We report for features which achieved significant classification performance or consistency across all the evaluation flies and discovery flies.

### Poor classification generalization due to heterogeneous feature value ranges among flies and datasets

We first checked whether core assumptions required to evaluate classification generalization were met. Specifically, to achieve good classification performance in the evaluation flies, we required feature value ranges in the evaluation flies to be similar to those of the discovery flies. Fig 4A shows the distributions of channel-averaged feature values across each of the flies from all the datasets, for the main wake and anesthesia/sleep conditions (see Methods). Visual inspection indicated high inter-fly variability in feature values, as well as heterogeneity in feature ranges across datasets. Specifically, flies from part of the multi-dosage and sleep datasets seemed to have feature values whose ranges differed greatly to those in the discovery flies (horizontal bands in Fig 4A).

Given the high variability across flies and datasets, we suspected that performance of the classifiers trained on the discovery flies would likely be low in the evaluation flies. The thin brown lines in Fig 4B show the number of significantly classifying features at each channel for each of the datasets. Across each of the evaluation datasets, the number of significantly performing features varied substantially (thin red lines in Fig 4B). In general, the number of significant features was greater in the multi-dosage evaluation flies (for MD8, 1,255–4,014 significantly performing features, $M = 2,912$, $SD = 780$ across 15 channels; and for MD4, 698–2,934, $M = 1,667$, $SD = 630$), slightly less in the single-dosage evaluation flies (0–2,304 significantly performing features, $M = 745$, $SD = 763$) and much less in the sleep evaluation flies (0–1,552 significantly performing features, $M = 260$, $SD = 397$). The other thin lines illustrate how the number of significantly performing features drops as the requirement for extends from classifying significantly in one dataset to two datasets, etc. Overall, consistent with our expectation, most features did not in fact generalize across all the datasets. In fact, only 47 features achieved significant classification performance across all the datasets, with all of them achieving so only at the deepest channel, Channel 1 (thick, dark blue line in Fig 4B).

Given that we only found significant features at Channel 1, it follows that the high-performing features we previously highlighted in the Stage 1 Pilot results, the stationarity-related feature `StatAvl250` and low-frequency power-related feature `SP_Summaries_welch_rect_logarea_2_1` at Channel 6, failed to generalize to all the evaluation flies. Across all the datasets, they achieved an average classification accuracy, weighted by number of flies in each set, of 55% and 51% at Channel 6, failing to achieve significant performance in the sleep flies (in the case of `StatAvl250`) or in the sleep and

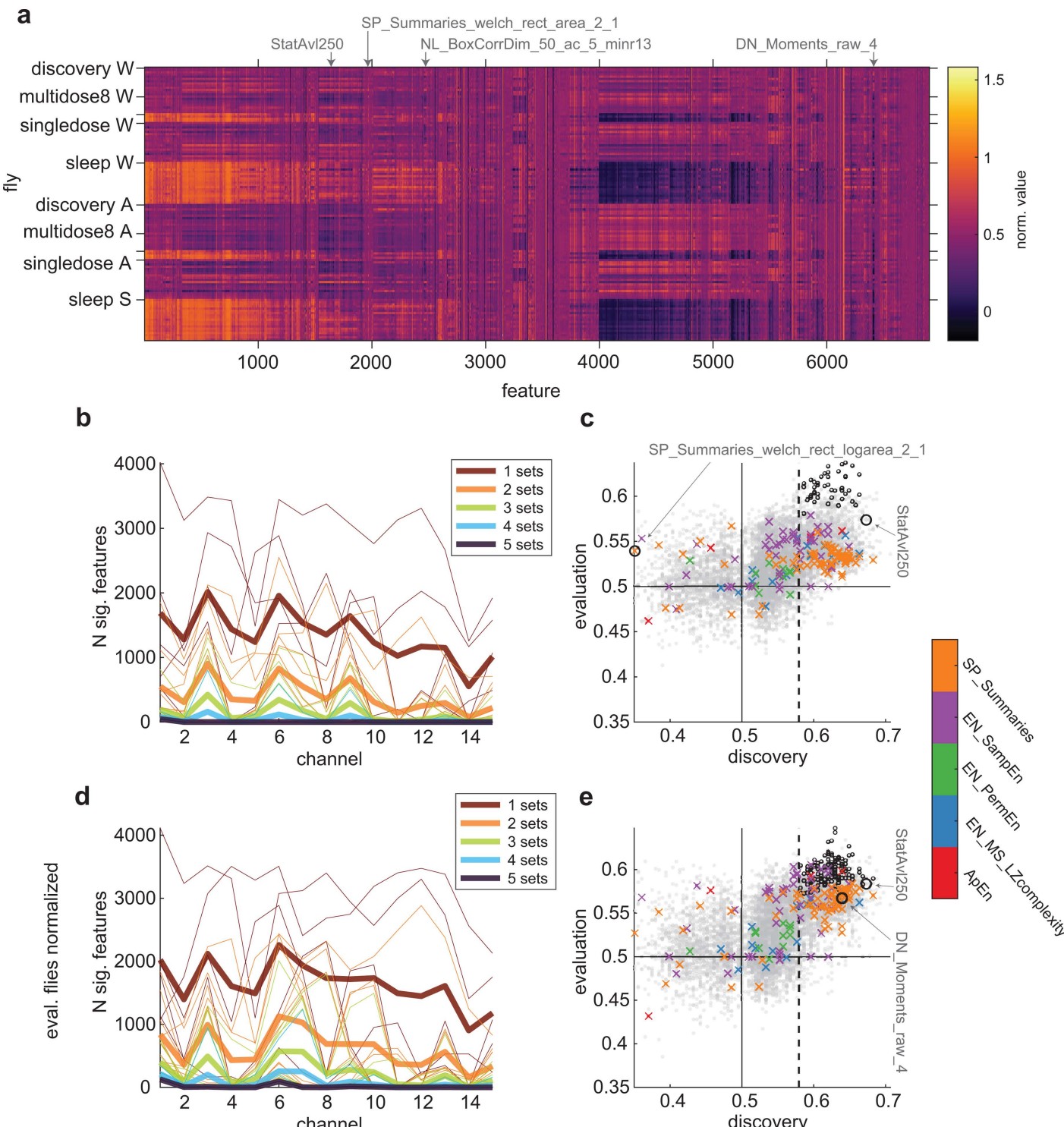

**Fig 4. Classification performance of *hctsa* features across all the flies. ( a)** Scaled values of *hctsa* features in all flies, averaged across epochs and channels. Each row corresponds to one fly from each of the datasets (*N* = 13 discovery, *N* = 8 MD8, *N* = 4 MD4, *N* = 18 single-dosage, and *N* = 19 sleep flies, respectively). Features (columns) are ordered based on hierarchical clustering using correlation (across time-series) distance between features. Arrows indicate features which attained the highest average classification performances across the evaluation flies. **(b)** Number of features which achieved statistically significant performance at each channel, in 1, 2, 3, 4, and 5 datasets, including the discovery flies, are indicated by the color of

lines as brown, orange, green, light blue and dark blue, respectively. Thin lines show for individual combinations of datasets, while thick lines show the average across all combinations. **(c)** Correlation of classification performances in the 13 discovery flies (x-axis) and average classification performances in the 49 evaluation flies (y-axis), at Channel 1. Each dot represents the classification performance of each of the features shown in **a**). Dots outlined with small circles indicate features which consistently achieved significant classification performance in all the flies. Solid horizontal and vertical lines indicate chance performance (= 0.5). Dashed vertical line indicates threshold for statistically significant across-fly classification performance in the discovery flies. Colored x's indicate performance of features related to previously described measures of conscious level (features related to spectral power across frequency bands and spectral edge frequency, `SP_Summaries` (orange); sample entropy, `EN_SampEn` (purple); permutation entropy, `EN_PermEn` (green); Lempel-Ziv complexity, `EN_MS_LZcomplexity` (blue); approximate entropy; `ApEn` (red)). **(d–e)** Same as **(b)** and **(c)**, but for classification after normalizing feature values in each set of evaluation flies to match the distribution of values in the discovery flies.

single-dosage flies (in the case of `SP_Summaries_welch_rect_logarea_2_1`). Further, the variance-related features `rms` and `standard_deviation` at Channel 5, which we previously highlighted as the best-performing features across all channels, both failed to generalize and did not achieve significant performance across all the datasets at any channels, including at Channel 1. At Channel 5, they only achieved significant performance in four of the multi-dosage evaluation flies (MD4; see Methods). The failure of these features in generalizing across all the evaluation flies illustrates the issue of focusing on the highest-performing features in a given dataset, which can fail to generalize to new datasets due to inter-dataset variability.

Meanwhile, the features which did achieve significant classification performance for all datasets achieved lower accuracies than those we highlighted in the Pilot results. Average classification performance, weighted by the number of flies in each dataset, ranged from roughly 60% to 64%. To illustrate how these features achieve significant classification performance across all the flies, but not others (such as those features highlighted in the Pilot results), we show the distribution of raw values for two features in Fig 5. Fig 5A–5C show the distributions of values for one significantly-performing feature, along with one feature which failed to generalize across all the datasets, the low-frequency power feature `SP_Summaries_fft_area_5_1`. While the distribution of values for the low-frequency power is such that there is an apparent difference in wake versus anesthesia/sleep values, the range of feature values is different to the discovery flies. As a result, across-fly classifiers trained on the discovery flies fail to classify the level of consciousness for these flies.

Fig 6A and 6B show each of the features which achieved significant performance in all datasets, their values for each of the flies and conditions, and how they cluster together based on correlating their values across all flies and epochs. There appeared to be several themes of features which significantly generalized, such as autocorrelation features (e.g., `AC_29`), fractal dimension related features (e.g., `NL_BoxCorrDim_50_ac_5_minr13`), and features related to outlier detection (e.g., `ST_LocalExtrema_n50_stdmax`). Despite the seeming variety in the themes of the features, feature values were highly correlated across epochs. As such, these features are likely capturing some similar common aspect of the time series in order to distinguish wake from anesthesia and sleep. In particular, we highlight the autocorrelation features (`AC_29` through `AC_34`), which are the simplest to understand. Specifically, these features indicate that there is higher autocorrelation in the time series during wakefulness at a timescale of roughly 30 ms (Fig 5D).

## Normalizing feature values only marginally improves generalization

Most features likely failed to generalize due to the different ranges of feature values across the datasets as seen previously in Figs 4 and 5. Specifically, the ranges of values for many features in the MD4 and sleep flies did not overlap with ranges of values for the discovery, MD8, or single-dosage flies, across both wake and unconscious conditions. For such cases, classifier performance would achieve only chance accuracy even if the wake and unconscious conditions are separable within the specific dataset. To account for this inter-dataset variation, we repeated the classification analysis after normalizing feature values in the evaluation flies such that the means and standard deviations across epochs matched those of the discovery flies (see Methods). In the context of a potential future marker in clinical settings, this normalization can be considered as corresponding to calibration of measurement devices to particular environments.

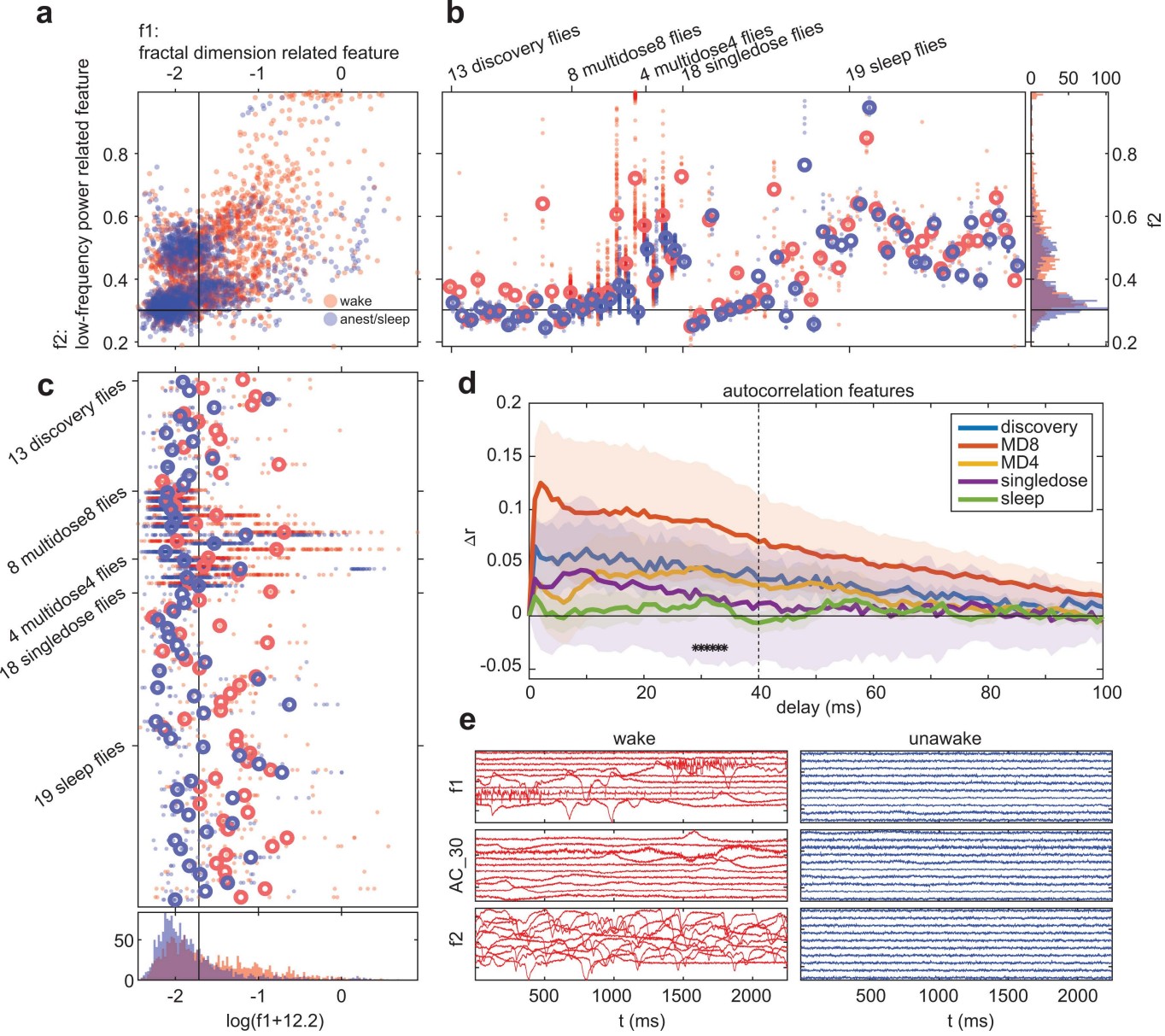

**Fig 5. Distributions of feature values in generalizing and non-generalizing features. ( a)** Distributions of values for a feature which achieved significant classification accuracy across all datasets (f1, `NL_BoxCorrDim_50_ac_5_minr13`, related to the concept of a fractal dimension, with the greatest weighted average performance across datasets, 64%) against one which achieved significant classification accuracy in the discovery flies, but not subsequently in all the evaluation flies (f2, `SP_Summaries_fft_area_5_1`, related to low-frequency power, with weighted average performance of 53%), both at Channel 1. Solid lines indicate discrimination thresholds obtained from the discovery flies. **(b, c)** Distributions for f1 and f2, grouped by individual flies. Open, bolded circles indicate median values for each fly. **(d)** Difference in autocorrelation features for each dataset (wake minus anesthesia or sleep, and averaged across flies, excluding 3 single-dosage evaluation flies, see Fig D of S1 Text). Vertical dotted line indicates the greatest time-delay for which autocorrelation was evaluated (in the *hctsa* feature set). Shaded areas indicate standard error across flies for each dataset. **(e)** Example time series for the epochs with the 10 greatest feature values across all epochs and flies for f1, autocorrelation at 30 ms (`AC_30`), and f2 during wakefulness (red), and the 10 smallest feature values during anesthesia/sleep (blue).

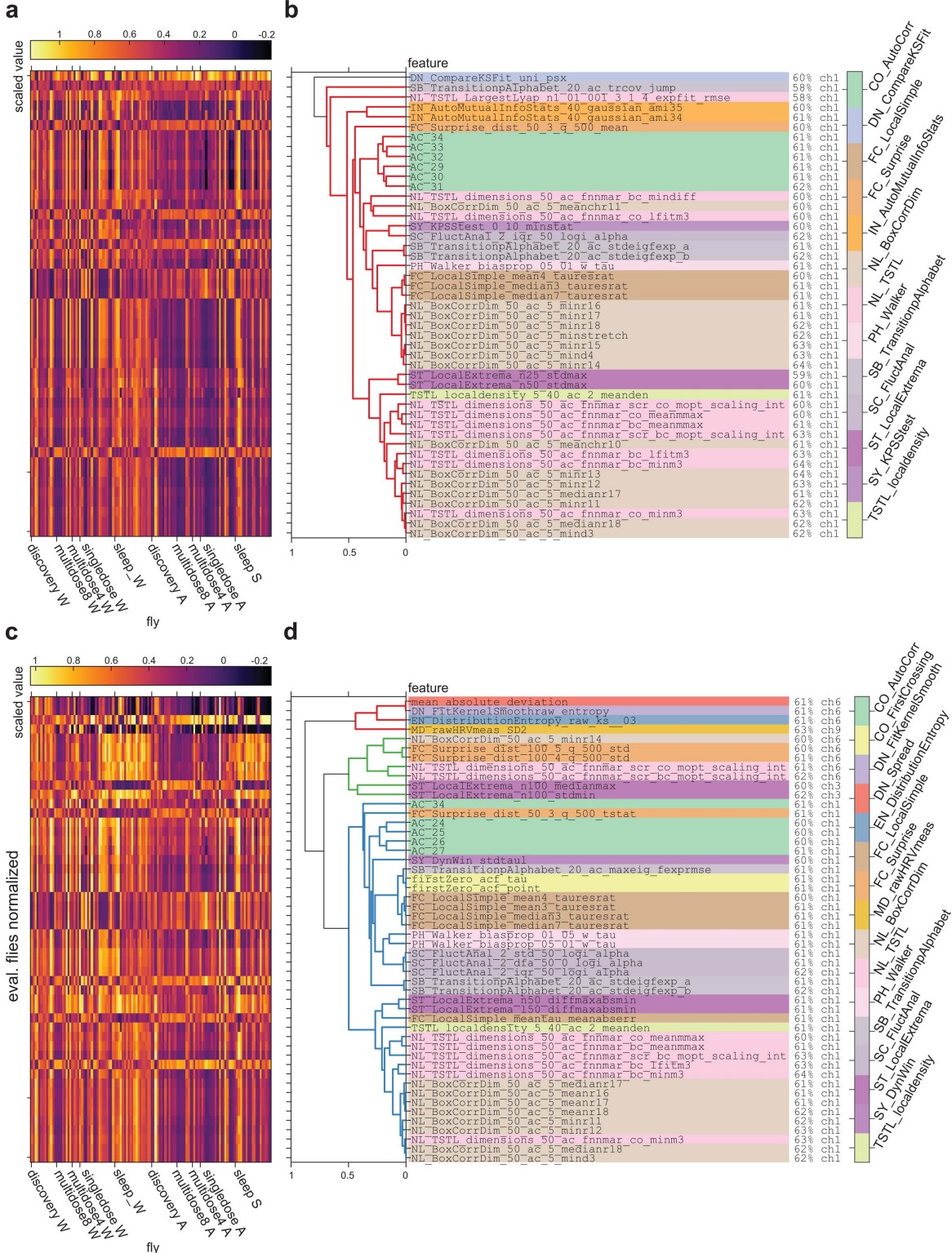

**Fig 6. Empirical grouping of significantly classifying features with and without normalizing evaluation flies. ( a)** Average feature values (across epochs and channels) from each fly, for each feature which consistently achieved significant classification performance across all the flies. Note that axes are swapped, compared to Fig 4A. **(b)** Agglomerative hierarchical cluster tree grouping significant features, constructed using Spearman correlation

distances as linkage distances between every pair of features (computed using all epochs from all flies). Dendrogram colors indicate groupings of features, using a threshold absolute Spearman correlation distance of 0.7. Individual feature names are listed to the right, along with their average classification performance in the evaluation flies. Colors indicate the broad theoretical category from which individual features belong (descriptions of each category can be found in the documentation for *hctsa*, at https://time-series-features.gitbook.io/hctsa-manual/information-about-hctsa/list-of-included-code-files). **(c, d)** Same as **(a)** and **(b)**, but showing for the top 50 features with the greatest average classification performance, across all channels, after normalizing feature values in each set of evaluation flies to match the distribution of values in the discovery flies.

Normalizing feature values in this manner marginally reduced the number of significantly classifying features in the MD8 flies (1,101–4,111 significantly performing features, $M = 2,823$, $SD = 823$ across 15 channels). Meanwhile, the number of significantly performing features greatly increased in the MD4 and sleep flies (1,066–3,507, $M = 2,811$, $SD = 727$ in the MD4 flies; and 0–2,095, $M = 892$, $SD = 587$ in the sleep flies), and marginally so in the single-dosage flies (0–2,319, $M = 754$, $SD = 775$). As such, normalizing feature values before performing classification most benefited the MD4 and sleep flies, consistent with the distributions of values shown in Figs 4A and 5B.

Surprisingly, when considering only features which achieved significant classification performance for all datasets, this normalization resulted in only a slightly higher number of significantly performing features. Fig 4E shows how while the classification performance for many features increased as a result of performing the normalization, most still failed to generalize. In total across the 15 channels, 263 features achieved significant classification accuracy across all the datasets. Channel 1 showed the greatest number of significantly performing features, with almost half of these features, 126, followed by Channel 6 with around a third, 92 (Fig 4D). Similar to the previous results without normalization, the highest performing feature which we highlighted in the pilot evaluation flies, kurtosis (`DN_Moments_raw_4`) previously in Channel 5, failed to generalize across all the datasets, at any channel.

Considering the 50 features that achieved the greatest classification performance at any of the channels (treating the same feature at different channels as separate), we found three general clusters of features (Fig 6D). One of these clusters (colored blue in Fig 6D) included only features at Channel 1 and corresponded to the autocorrelation-related features which previously generalized when performing classification without normalization (cf. Fig 6A and 6B). The other two clusters included features from different channels (specifically Channels 3, 6, and 9). The first of these (colored green in Fig 6D) included features similar to the features highlighted previously for Channel 1, again related to fractal dimension and variation in extreme values in short time windows, but at mainly Channel 6. The separation of this cluster from the previous suggests some small difference in the autocorrelation of signals from the center of the fly brain with those obtained elsewhere. The second of these additional clusters (colored red in Fig 6D) included features related to variability in time samples, similar to those highlighted in our consistency analysis of the Pilot results, such as `MD_rawHRVmeas_SD2`. We provide classification performances for all features and channels in https://osf.io/8wvsq/?view_only=8a056d1c573b4f23a6cf6cea8b976ddb.

## Consistent effect of loss of consciousness even in features that did not exhibit significant classification ability

While the above normalization matched the distributions of feature values across datasets, it did not address variability among individual flies. Specifically, while the ranges of feature values may have varied greatly among flies, the direction of the effect of anesthesia or sleep may have been consistent across flies. So, to address the high inter-fly variability in feature values, we next ignored individual differences in raw feature values and evaluated the degree to which within-fly effect direction consistencies generalized to the full set of evaluation flies. That is, we aimed to investigate whether some features reliably increase (or decrease) from wake to anesthesia and sleep, making them a reliable relative indicator of a change in conscious arousal within a given fly.

Removing individual differences in this manner yielded slightly fewer significant features than for classification in the MD8 flies (208–3,788, $M = 2,252$, $SD = 975$ across 15 channels), but many more for the other flies (1,763–3,741, $M = 2,857$,

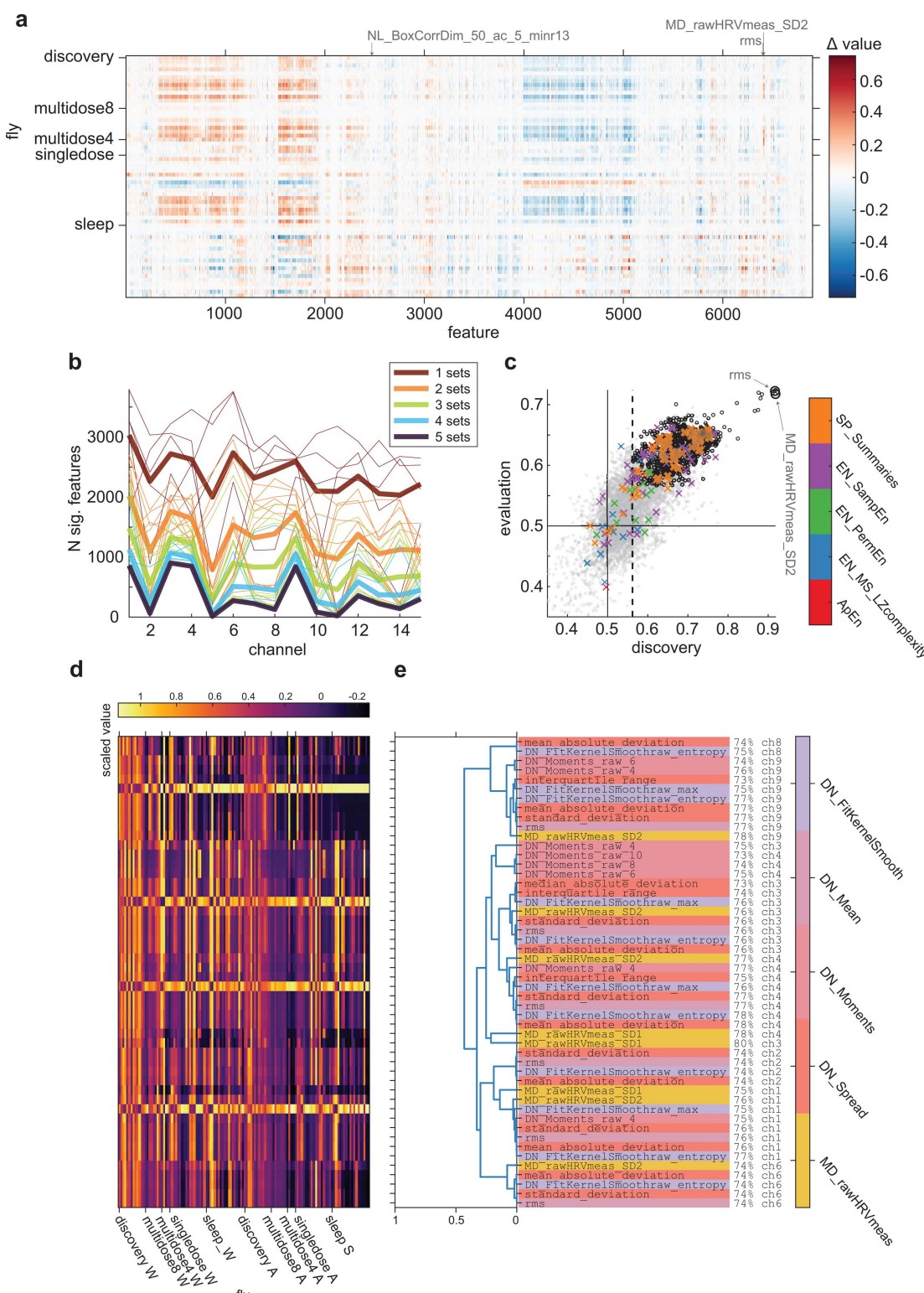

**Fig 7. Within-fly effect direction (wake – anesthesia) consistency of *hctsa* features across all flies. ( a)** Differences in scaled *hctsa* values between wakefulness and anesthesia/sleep in all the flies, averaged across channels and epochs for each feature. Each row corresponds to one fly from each of the datasets (*N* = 13 discovery, *N* = 8 MD8, *N* = 4 MD4, *N* = 18 single-dosage, and *N* = 19 sleep flies respectively). Features (columns) are

ordered as in Fig 4A. **(b)** Number of features which achieved statistically significant within-fly direction effect consistency at each channel, in 1, 2, 3, 4, and 5 datasets, including the discovery flies, are indicated by the color of lines as brown, orange, green, light blue and dark blue, respectively. Thin lines show for individual combinations of datasets, while thick lines show the average across all combinations. **(c)** Correlation of classification performances in the 13 discovery flies (x-axis) and average classification performances in the 49 evaluation flies (y-axis). Each dot represents the classification performance of each of the features shown in a). Dots outlined with small circles indicate features which achieved significant consistency in all the datasets. Solid horizontal and vertical lines indicate chance performance (= 0.5). Dashed vertical line indicates threshold for statistically significant consistency in the discovery flies. Colored x's indicate performances of features related to previously described measures of conscious level, as described in Fig 4A and indicated by the color bar. Thicker x's indicate those features which achieved significant consistency in all the datasets. **(d)** Average feature values (across epochs and channels) from each fly, for the 50 features with the greatest consistencies across all the flies, across all 15 channels. **(e)** Agglomerative hierarchical cluster tree grouping significant features, constructed using Spearman correlation distances as linkage distances between every pair of features (computed using all epochs from all flies). Dendrogram colors indicate groupings of features, using a threshold absolute Spearman correlation distance of 0.7. Individual feature names are listed to the right, along with their average classification performance in the evaluation flies. Colors indicate the broad theoretical category from which individual features belong.

$SD = 480$ in the MD4 flies; 70–3,063, $M = 1,846$, $SD = 1,009$ in the single-dosage flies; and 649–2028, $M = 1,728$, $SD = 577$ in the sleep flies). Features may achieve significant classification accuracy but not consistency in cases where there is greater variability in feature values during, e.g., wakefulness, such that they surround the values for anesthesia. In such a scenario, depending on the trained classifier threshold, a large majority of anesthesia data points may be classified correctly, along with a minority of wakeful data points, leading to significant classification performance but not consistency.

In focusing on the direction of the effect of anesthesia and sleep within flies, many features appeared to be able to discriminate between wake and anesthesia or sleep. Fig 7A shows differences in feature values, at Channel 1, between wake and unconscious epochs to illustrate within-fly effect direction consistency for each feature across all the flies. In contrast to raw values (Fig 4A), which showed greater variability in feature values among flies than among conscious levels, the direction of the effect of anesthesia or sleep for many features appeared to be reliable across flies for many features. This is consistent with what we previously reported in the Pilot results, where the direction of the effect of anesthesia seemed consistent across flies even despite mediocre classification performance.

As in the Pilot results, there again appeared to be clusters of features (such as in columns ~2,900 to ~3,000, or columns ~3,600 to ~3,800) for which consistency would be high. However, the number of features in these columns was an order of magnitude smaller than what we previously reported in the Pilot results. Specifically, when considering only features for which the direction of the effect of anesthesia or sleep was significantly consistent, the number of significant features ranged from 13 to 902 across the 15 channels, with the greatest number of significant features occurring at Channel 3 (Fig 7B). This is likely a consequence of variability across datasets from factors such as differences in exact experimental setup (e.g., electrode location), severely limiting inter-fly generalization.

Fig 7C shows how within-fly effect direction consistency at Channel 1 generalized to the evaluation flies. In contrast to our classification analysis, features related to well-known markers of consciousness, i.e., those related to spectral power and complexity, showed significant within-fly effect direction consistency (thicker x's in Fig 7C). However, these did not achieve the greatest averaged consistencies across the datasets. Instead, features related to signal variance generally achieved the greatest consistencies. The feature which achieved the greatest consistency was `MD_rawHRVmeas_SD1` at Channel 3, a measure of variability between consecutive time samples (conceptually related to `MD_rawHRVmeas_SD2` which we previously highlighted in the Pilot results, which also achieved a significant consistency of 0.73 at Channel 3; both features also achieved significance in Channel 6, 0.69, and 0.68, respectively for `SD1` and `SD2`). The top 50 features, taken from any of the 15 channels, were all related to descriptions of the distribution of time samples (such as standard deviation, `standard_deviation`, and distributional moments, `DN_Moments_raw`), and clustering of these indicated that their pattern of values across epochs were all indeed highly correlated, even across channels (Fig 7E). These results suggest that relatively simple measures related to variability and distributional shape may be more reliable in distinguishing conscious levels in individuals than, e.g., low-frequency power or complexity measures, when baseline

measurements are available. We provide within-fly effect direction consistencies for all features and channels in https://osf.io/8wvsq/?view_only=8a056d1c573b4f23a6cf6cea8b976ddb.

## Discussion

In this registered report, we used the *hctsa* univariate time-series feature library, the most comprehensive set of available univariate time-series features, to search for features which may serve as potential markers of consciousness. In our Stage 1 manuscript, we first searched for features which discriminate wakefulness from isoflurane anesthesia in a discovery set of flies ($N = 13$), training and fixing classifiers for each feature using only these flies. To account for the potential for poor generalization, we also evaluated the degree to which the direction of the effect of anesthesia or sleep on feature values was consistent among these flies.

In Stage 2, we evaluated the degree to which the classifiers trained in Stage 1 discriminated wakefulness from loss of consciousness in independent sets of evaluation flies, including conditions of anesthesia ($N = 28$) and sleep ($N = 19$), which were previously blinded to the data analysis team. Here, we found only a small set of features to achieve significant performance across all the datasets, at the center of the fly brain (Channel 1). We also evaluated the within-fly effect direction consistency for all the flies, finding many more features to achieve significant consistency than for classification. These features included those related to well known, previously reported markers of consciousness, such as spectral power or complexity measures, though features with the highest consistencies were related to measures of variance.

Across-subject classification using single, specific features—which is ideal for a consciousness measure—is rarely evaluated across independent datasets involving new unseen levels of consciousness, especially in the context of "discovering" and proposing new potential consciousness markers. Studies which do evaluate already-proposed markers across independent datasets consisting the same conscious conditions, such as [14], still report limited generalization performance, likely because of the strong focus on traditional spectral power or complexity measures which may be more suited for within-subject comparisons. We elaborate on these points below.

### Within-dataset generalization versus blinded generalization to new datasets

We originally focused on Channel 6 in our Stage 1 Pilot results as it showed the greatest number of significantly performing features in the discovery flies. Further, many of these features related to prominent consciousness markers, such as those related to low-frequency spectral power, achieved above chance cross-validated classification performance in these flies. While many of these already failed to generalize to the pilot evaluation flies, one feature related to low-frequency spectral power did achieve significant generalization, in fact achieving the greatest classification performance in the pilot evaluation flies. We also highlighted better-performing features, related to variance of voltage fluctuations and stationarity. Normalizing feature values in the pilot evaluation flies to match the means and standard deviations in the discovery flies allowed many of these features to achieve significant classification accuracy.

However, none of these previously highlighted features generalized across all four sets of evaluation flies, even when normalizing feature values to match the means and standard deviations of those in the discovery flies. Instead, we identified features which previously were overshadowed in terms of classification performance. The failure of previously highlighted features in generalizing to the evaluation flies illustrates potential drawbacks of evaluating individual candidate consciousness markers in single datasets, even when carrying out cross-validation. Specifically, focusing on single individual datasets inflates the seeming performance of particular features (whereas across datasets, there may be high variation in ranges of feature values, especially in cases where inter-experimental variability may be substantial), while focusing on particular candidate markers greatly limits the potential for discovering other candidate markers. These drawbacks are problematic as it can lead to frequent supposed discoveries of new consciousness markers which ultimately do not generalize. More generally, selection of specific analysis methods by different research groups, even on the same

dataset, can lead to inconsistent conclusions [55]. These issues apply equally to our own previous publication [17], which requires further validation studies with larger, unseen datasets in order to better evaluate generalization.

One concrete factor which contributed to the lack of generalization for many features was the difference in feature value ranges across datasets and individual flies. In particular, values in the MD4 and sleep flies tended to be overall much greater or much smaller than in the other flies. For example, for a feature related to low frequency power, almost all the values in these flies were above the threshold for classifying wake from anesthesia trained in the discovery flies. If this was the case for just the sleep dataset, this might be interpreted as sleeping flies having a higher level of consciousness compared to anesthetized flies. However, a same difference in value ranges was present also in the MD4 flies. The difference in feature value ranges might also be attributed to variations in recording setup. However, all the datasets analyzed here were recorded from the same laboratory (BvS), and all the evaluation datasets were recorded by one PhD student (TJ) using the same recording setups. As such, while there is likely variation in exact channel location across all the flies (Fig 1F from [43] illustrates this variability for a whole-brain preparation), systematic differences in channel location across the datasets is unlikely to be the case (see S1 Text regarding difference in depth of electrode probe insertion between the discovery and all the evaluation flies). This is especially the case across the evaluation datasets due to a method for consistent electrode placement ([46]; which was developed after recordings from the discovery flies were made). Other potential sources of variation might lie in seemingly small experimental differences, such as the age of the flies (see Methods).

### Why do so many features fail to generalize despite consistent direction of effect of loss of wakefulness?

We evaluated two forms of generalization. As discussed above, we first evaluated the degree to which the classifiers trained on the discovery flies could predict wakefulness from loss of consciousness in the evaluation flies. We next evaluated the degree to which the direction of the effect of anesthesia or sleep in the evaluation flies was consistent with that in the discovery flies. In principle, evaluating within-fly effect direction consistency factors out inter-individual and inter-dataset differences in feature value ranges. While we found few features to achieve significant classification performance, we found many more features to have a consistent direction of effect of anesthesia or sleep. As such, much of the low generalizability in classification performance was likely to have been due to differences in feature value ranges, rather than, e.g., inability to separate wake from anesthesia or sleep in some datasets. This suggests that a potential marker of consciousness may globally shift across datasets. As potential reasons for this global shift, we consider two possible explanations.

The first explanation relates to variation in experimental and recording setup as we raised earlier. In addition to variation in exact electrode placement and age of the flies, other differences which may have led to shifts in values include obtaining recordings in a dark versus lit-up room, presentation of stimuli relevant to the original experiments in between periods which we analyze, such as flickering lights or air puffs, and so on. However, we did not expect these factors to greatly affect a potential marker of conscious level, especially if we assumed flies to have the same level of consciousness in all wake conditions and again in all the anesthetized and sleep conditions. If this is the case, then features which showed significant consistency do not generalize enough to be considered ideal consciousness markers which can be applied across many contexts. Rather, they require some baseline observation made with the same setup and in the same environment to be able to inform as to an individual's consciousness level. Or conversely, conditions during a test observation should be manipulated to be comparable to those during a baseline observation. Which specific environmental factors are more important for this may be clarified through experiments explicitly manipulating the background environment.

The second explanation is sensitivity to inter-individual differences. The sleep evaluation flies consisted of flies which were several days younger than in the other datasets. However, we consider this explanation unlikely, as the MD4 flies who also appeared to have shifted value ranges, share the same age range as the other multi-dosage evaluation flies. However, baseline levels of consciousness may have varied among flies due to individual sensitivity and recovery to electrode insertion and the accompanying cold anesthesia during preparation, or sleepiness due to time of the recording. The

problem of uncertain baseline levels of consciousness introduces a level of circularity in evaluating potential conscious-ness markers, but is somewhat mitigated through the registered report format where authors and reviewers agree on and fix a definition of conscious/non-conscious (and its implied variability across individuals), before evaluating candidate markers. In this vein, we are currently also conducting another registered report study to evaluate consistency of candi-date markers between human and monkey neurophysiology data [56].

Lastly, we acknowledge an important assumption in this study—that the neural process(es) underlying consciousness are affected in the same way during loss of consciousness for both anesthesia and sleep. However, there are clear differ-ences between anesthesia and sleep, such as the possible presence of dreams during sleep in humans and maybe even flies [46], or the capability of human subjects to follow verbal commands during various sleep stages [57]. As such, the general lack of generalization across most features may also be interpreted in several ways. One simple interpretation is that sleep does not induce a loss of consciousness to the same degree as anesthesia, such that it presents as an inter-mediate level of consciousness between anesthesia and wakefulness (e.g., in the case of dreams). Another is that neither anesthesia nor sleep entail the complete loss of consciousness, and each follows a different path of breaking down of the process(es) underlying consciousness. A third interpretation is that non-conscious neural processes during sleep such as those related to memory consolidation, which are suppressed under anesthesia, may be masking the breakdown of consciousness related processes. Given this interpretation, many of the univariate features in *hctsa* lack the sensitivity to distinguish these processes from consciousness related ones. In this vein, more complex bi- or multi-variate features which incorporate information across the brain might be more able to distinguish such processes.

## The importance of investigating individual features

Here, we evaluated and compared the performance of individual features available in *hctsa* in distinguishing between wake and anesthesia or sleep. In doing so, it is possible to identify specific time-series features or overall themes which similar features capture. This may inspire new hypotheses about how neural activity leads to consciousness. Meanwhile, it is also possible to combine multiple features together with multivariate classification. In particular, combining features in this manner may achieve greater classification performance in distinguishing conscious levels. There is already work following this approach, combining already proposed markers of conscious level in the literature [13,14] to achieve better classification of disorders of consciousness. This avenue is particularly appealing due to the wide range of analysis types available in *hctsa*. However, this kind of analysis has drawbacks regarding interpretation of features, and should be car-ried out carefully.

Generally speaking, multivariate classifiers require fitting large numbers of parameters. However, when the amount of parameters to fit exceeds the available training data, these classifiers can severely suffer from overfitting. Most published studies claim to overcome this difficulty using the process of "cross-validation", which repeatedly uses different "folds" of a given dataset for training and testing (which we conducted in the discovery flies). However, cross-validation only addresses this problem if the data is sufficiently varied to represent the true variability in the full, global population. While even an overfitted model can provide good generalization to new data which occurs with the "region" of the training data [58], classification can suffer greatly in cases of data outside those regions, e.g., the addition of a sleep condition in our evaluation flies. Meanwhile, univariate analysis requires much less parameter fitting, drastically reducing the chance of overfitting. Even so, we already observe something related already in our univariate classification, where classification suf-fers greatly from the inclusion of a sleep dataset, when the classifiers were trained on an anesthesia dataset.

A more important issue regards the interpretability of multivariate classifiers. While combining multiple features into a single classifier can greatly improve classification performance (ignoring the issue of overfitting), how to interpret a bundle of wildly varying candidate consciousness markers beyond evaluating their contribution in improving classification per-formance (such as in, e.g., [14]) is unclear. This is the case in recent, successful artificial neural networks such as large language models, where huge numbers of parameters are fit to extremely large and varied datasets. Such models are

able to provide seemingly good predictions, but in a completely black-box manner, where the mechanism by which these predictions are generated is abstracted out. Thus, from an application viewpoint, better classification performance using multivariate classifiers can be appealing. However, from a scientific viewpoint, pursuit of better classification performance may not necessarily deepen our understanding of a given phenomenon. To pursue the latter, we now turn to interpreting the univariate features we highlighted in our results.

## Autocorrelation as a marker of conscious level

In focusing on individual features, we identify a theme of analysis—autocorrelation—which to our knowledge has not received much attention in consciousness research. Interestingly, features related to the concept of spectral power did not successfully generalize across all the datasets, despite autocorrelation and the power spectrum being directly related through a Fourier transform. This may, however, be due to lack of granularity of the available features in *hctsa* covering the frequency domain, compared to the available autocorrelation features.

Many of the features which significantly discriminate wake from anesthesia and sleep, are related to the notion of fractal dimension. However, these features (and in general, features which include "`_ac`" in their name) include early processing steps which depend on properties of autocorrelation. For example, `NL_BoxCorrDim_50_ac_5_minr13` first generates a time-delay embedding based on when the autocorrelation function first reaches zero. Despite including more complicated later processing steps, these fractal dimension features achieved comparable discrimination performance to the simpler autocorrelation features. Given these points, and the in general strong correlations among significantly classifying features, it is likely that the performances of such fractal dimension features are in fact being driven by autocorrelation properties.

The concept of fractal dimensions has been somewhat explored in the context of detecting drowsiness and discriminating sleep stages in EEG recordings. Specifically, measures of fractal dimension have been reported to be reduced during drowsiness [59,60] and in deeper sleep stages [61,62], and during anesthesia [63,64], consistent with our own results. However, the theoretical motivation thus far of using such a method has been limited to using a complex nonlinear analysis in the hopes of capturing so-called complex nonlinear dynamics in the brain. Given our findings, and lack of a priori theoretical motivation for investigating fractal dimension related measures, results from these studies may simply be reflecting differences in autocorrelation.

## Directions for future investigation

We have evaluated a vast library of time-series features as candidate markers of conscious level, and identified features which generalize across multiple independent fly datasets. In doing so, we are taking steps similar to an "iterative natural kind" approach to finding consciousness markers [65]. In this approach, we assume consciousness to be a "natural kind", that is, that conscious systems share an underlying nature which is identifiable through iterative procedures. Specifically, we make the assumption that a relatively simple system such as the fly brain is conscious to identify potential pre-theoretical markers of consciousness which should then be gradually tested in more complex systems and developed into a theory whose explanatory power and simplicity can be evaluated. This in contrast to starting from pre-established or popular markers (such as applying power spectral measures which seem to discriminate conscious levels in human recordings in order to test if they are conscious), or to starting from some kind of theory and testing its predictions.

While there are several popular theories of consciousness [66,67], one stands out in providing an operationalized measure which can be directly applied to neural recordings in general. Integrated information theory [68–71] attempts to start from first principles, identifying universal aspects of consciousness and then deriving a multivariate measure, integrated information, which reflects the extent to which a system supports these aspects. We previously applied one version of integrated information [70] to the discovery flies, finding similar classification performance as the high performing features in the Pilot results (i.e., higher performance than what we reported in the Stage 2 Results). However, we did not evaluate generalization to new datasets as we did in the present study. Further, as a multivariate measure, computed across

multiple channels, it is not clear whether a direct, fair comparison can be made between integrated information and the univariate features evaluated here.

A more fair comparison can be made, however, equating the number of channels used, with other bi- or multi-variate measures. In this regard, there is a more recent toolbox, *pyspi* [72], which provides over 250 bivariate analysis methods, such as correlation, coherence, and Granger causality, and includes the theory-driven integrated information. Given that a common idea is that it is the interactions among neurons or brain regions which matter for consciousness, evaluating whether these multivariate features perform better and generalize more successfully than the univariate features in *hctsa*—or whether combining multivariate and univariate features improves generalization beyond using either alone [73]—is a clear next step to take in finding and evaluating potential consciousness markers.

Overall, this work highlights the limitations of the standard cross-validation approach in discovering and validation potential measures of consciousness. In particular, we show that while standard cross-validation within a dataset can find potential markers, they may not generalize to independently obtained datasets. Further exploration of properties relating to autocorrelation may lead to a reliable across-subject consciousness marker utilizing some regularized combination of fractal dimension-related features. Understanding if and how such features are related to existing consciousness theories will help distinguish among them. Meanwhile, the extension of the exploration of features to include bi- or multi-variate measures characterizing interactions among neural populations, especially any which are proposed from theories of consciousness, will also be a fruitful avenue toward better identifying conscious level across subjects.

## Supporting information

**S1 Text. Correlations in feature values between discovery and evaluation flies with and without offsetting evaluation fly channels.** Supplementary figures.
(PDF)

## Author contributions

**Conceptualization:** Naotsugu Tsuchiya.

**Data curation:** Travis Jeans.

**Formal analysis:** Angus Leung, Ahmed Mahmoud.

**Funding acquisition:** Bruno van Swinderen, Naotsugu Tsuchiya.

**Investigation:** Angus Leung, Ahmed Mahmoud, Ben D. Fulcher.

**Methodology:** Angus Leung, Travis Jeans, Bruno van Swinderen, Naotsugu Tsuchiya.

**Resources:** Travis Jeans, Ben D. Fulcher, Bruno van Swinderen, Naotsugu Tsuchiya.

**Software:** Angus Leung, Ahmed Mahmoud.

**Supervision:** Ben D. Fulcher, Bruno van Swinderen, Naotsugu Tsuchiya.

**Visualization:** Angus Leung.

**Writing – original draft:** Angus Leung, Ahmed Mahmoud.

**Writing – review & editing:** Angus Leung, Travis Jeans, Ben D. Fulcher, Bruno van Swinderen, Naotsugu Tsuchiya.

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
