## [Editor Report · Decision Letter 0]

Dear Dr Leung,

Thank you for submitting your manuscript entitled "Towards blinded classification of levels of consciousness: distinguishing wakefulness from general anesthesia in flies using a massive library of univariate time series analyses" for consideration as a Preregistered Research Article by PLOS Biology. Please accept my apologies for the delay in sending the decision below to you.

Your manuscript has now been evaluated by the PLOS Biology editorial staff. We have also discussed your proposal with two academic editors, one with expertise in the biological question you are addressing and another one with expertise in Pre-registered Reports. I am writing to let you know that we are interested in peer-reviewing your proposal, but before we can commit to that, we would like you to revise your submission by including a summary table that aligns each research question with the hypothesis/es used to answer the question, the sampling plan for each hypothesis (e.g. power analysis, where applicable), the specific statistical analysis/es that will be used to test the hypothesis, and a pre-specification of which outcomes will confirm or disconfirm the hypothesis (to varying degrees of strength where multiple analyses with different possible outcomes are used to interrogate one hypothesis). You can read our guidelines for Pre-registered Reports here: https://plos-marketing.s3.amazonaws.com/Marketing/Biology+Preregistered+Articles+Guidelines+for+Authors.pdf

In addition, we need you to complete your submission by providing the metadata that is required for full assessment. To this end, please login to Editorial Manager where you will find the paper in the 'Submissions Needing Revisions' folder on your homepage. Please click 'Revise Submission' from the Action Links and complete all additional questions in the submission questionnaire.

Please re-submit your manuscript within two working days, i.e. by Oct 28 2021 11:59PM. Do let me know if you need more time or would like to discuss our decision.

Kind regards,

Gabriel

Gabriel Gasque

Senior Editor

PLOS Biology

ggasque@plos.org

---

## [Decision Letter · Decision Letter 1]

Dear Dr Leung,

Thank you for submitting your manuscript "Towards blinded classification of levels of consciousness: distinguishing wakefulness from general anesthesia in flies using a massive library of univariate time series analyses" for consideration as a Preregistered Research Article at PLOS Biology. Your manuscript has been evaluated by the PLOS Biology editors, by two academic editors with relevant expertise, and by four independent reviewers. Please accept my apologies for the long delay in sending the decision below to you.

In light of the reviews (below), we will not be able to accept the current version of the manuscript, but we would welcome re-submission of a much-revised version that takes into account the reviewers' comments. We cannot make any decision about publication until we have seen the revised manuscript and your response to the reviewers' comments. Your revised manuscript is also likely to be sent for further evaluation by the reviewers. We expect your revision to gather enough support from the reviewers for us to consider eventual acceptance.

One academic editor also provided specific feedback (below), which you should also address.

We expect to receive your revised manuscript within 3 months.

**IMPORTANT - SUBMITTING YOUR REVISION**

Your revisions should address the specific points made by each reviewer. Please be thorough when addressing reviewer 1's comments, as the concerns raised about conceptual advance are particularly important. Please submit the following files along with your revised manuscript:

*Re-submission Checklist*

*Published Peer Review*

*PLOS Data Policy*

Sincerely,

Gabriel Gasque

Senior Editor

PLOS Biology

ggasque@plos.org

REVIEWS:

Academic editor: I would invite the authors to specify better how they will evaluate different effects on different electrodes. What would it mean if a feature significantly predicts the difference between wakefulness and anesthesia in one channel (either within or across flies), but that same feature does not for another channel? How are the authors going to arbitrate between whether these differences are due to methodological/statistical factors (e.g. different signal-to-noise ratios in the different channels), which may be less interesting, or whether they relate to meaningful differences in the brain structures that are sampled?

If, as anticipated, many neural measures may predict the difference between wakefulness and anesthesia (even for one channel), the authors have to think about an appropriate way to summarize those results in an intuitive manner for the reader. How are the authors planning to do so? Are there specific “types” of measures that are specifically informative? Is there a way to cluster time-series features (see a similar comment by reviewer 4) and link those features back to the existing literature on the known neural processes underlying the change from wakefulness to anesthesia?

Reviewer #1: The neural mechanism of anesthesia action remains to be fully elucidated. Using previously recorded LFP signals of flies with/without anesthesia (Cohen et al. 2018 eNeuro), in addition to a newly collected dataset, Leung et al aim to develop a protocol for classifying the depth of anesthesia based on LFP characteristics. Identifying quantitative alterations in LFP signals in the context of anesthesia is quite valuable to a broader neuroscience community beyond the immediate field of anesthetics research. However, based on the authors study design, there are several conceptual and technical gaps that seem to preclude its potential utility to the broader scientific community. At the current stage of this report, this rather specialized study with a debatable conceptual leap in the definition of consciousness seems to be more suitable for a journal dedicated to computational neuroscience.

Major points:

1. It is not clear what (if any) broad biological insight this study by itself will provide. Previous studies by the van Swinderen group establish changes in LFP signals associated with a particular type of anesthetics, isoflurane. However, in this report, the researchers do not propose to compare the time series measures against established differences, nor do they seem to make an attempt to compare a certain subset of interesting features against established measures. Furthermore, as discussed in the manuscript, it is apparent that there will be hundreds of time series features that may be successfully extracted and generalized. What would be the essence of finding these many features? More importantly, it is not clear what identification of a particular set of features extracted from data obtained under a single experimental condition will tell us about either biology of anesthesia or a deeper meaning of "consciousness".

2. Dose-response effects. The authors propose to compare a single dose (0.6 % iso) against the no anesthesia condition. However, it is not clear how this dose was selected. Conceivably, a subset of LFP features could emerge at different "levels" of anesthesia. Using this study design, the authors could miss key features related to anesthetic induction, or may generalize effects which are specific to the selected dose. It would be important to explore/analyze the LFP during induction and recovery.

3. Variety of anesthetics. It is utmost important for the authors to demonstrate that the time series features of LFP attributable to "consciousness" (or lack of it) are independent of the types of anesthetics. Otherwise, there will be no ground for the proposal to equate the altered time-series features during anesthesia to a LFP component due to loss of consciousness.

4. The hctsa framework is not well-known to the general biology audience of the Journal. It would be helpful for the authors to better describe the library and to give specific examples of some univariate time-series features and spell out more concrete meaning and implications.

5. According to Fulcher et al., 2013, several questions can be raised regarding implementation.

1. Apart from line noise removal, were there any pre-processing/normalization of LFP signals? For several of the hctsa analyses, it would seem that pre-processing was necessary. For example, one would expect that the ST_propsimp function, which measures the proportion of positive vs. negative vs. zero values would be highly sensitive to the DC value of the signal. This measure could be very useful for DC-subtracted data, but meaningless if the DC offset is included.

2. Many measures include hard-coded parameters, e.g. CO_autocorr uses tau = 1 … 40. Are these parameters values appropriate? One would expect the interpretation of lag would be dependent on sampling rate. A 40-point lag on a 20 kHz signal is very different than that on a 1000 Hz signal.

3. How many of these measures require a signal to be stationary for proper interpretation? Is there evidence that the LFP signals during the anesthesia epochs are stationary?

6. Study power & sample size. The authors should justify why a sample size of 13 flies is sufficient for this analysis. Similarly, the test sample size of 12 should be justified. (Both seem somewhat low)

7. False discovery rate. A broader discussion of the Benjamini-Hochberg procedure is warranted as many of the tests in hctsa are clearly not independent to one-another. For example, CO_autocorr, t = 1 and CO_autocorr, t=2 would be expected to yield similar results.

Reviewer #2: Stage 1 review

The importance of the research question(s).

There are various theories for the neural basis of conciousness which suggest various signatures and mechanisms, however, there is little agreement. I believe the research question posed by the authors, which in short is trying to understand the fundamental mechanisms of conciousness, is important, both from the perspective of scientific understanding but also because it has clinical implications.

The specific research questions, 'What univariate time-series features (from hctsa) can serve as markers of level of consciousness ACROSS individuals?' (and WITHIN), are both important questions which would help us both predict (assess clinical patients for example) or understand mechanisms.

The logic, rationale, and plausibility of the proposed hypotheses (does the manuscript provide a valid rationale for the proposed study, with clearly identified and justified research questions?

The proposed hypotheses (testing each feature at each channel) are logical and rationale. The authors present a data-driven approach to identify time-series that are predictive above chance of conciousness. Until now, most research has been focussed on pre-selected features (either measures of spectral power, or within their own realm of expertise). Taking an indiscrimnate approach removes the feature selection bias. The hypotheses are entirely plausible given that predetermined features have shown some efficacy previously.

The soundness and feasibility of the methodology and analysis pipeline (including statistical power analysis where appropriate). Is the protocol technically sound and planned in a manner that will lead to a meaningful outcome and allow testing of the stated hypotheses?

As made clear by the authors, their methodology introduces a multiple hypothesis problem which the authors will correct for with (i) FDR, (ii) leave one out testing, and (iii) independent evaluation. The protocol would lead to results that a statistically robust. Feasability is clear with the existing hctsa package, pilot results and existing data.

Whether the clarity and degree of methodological detail is sufficient to exactly replicate the proposed experimental procedures and analysis pipeline.

Regarding experimental methodology - it seems clear to me, but I am not an experimentalist and therefore would rely upon other reviewers to check the clarity and detail of the experimental methodology.

The analysis methodology was detailed and clear, and could easily be replicated.

Whether the authors have pre-specified sufficient outcome-neutral tests for ensuring that the results obtained are able to test the stated hypotheses, including positive controls and quality checks.

The authors have clear stated tests that provide outcomes. The authors use an independent classification set, whose importance becomes evident in the pilot results section.

Additional review notes.

The WITHIN fly analysis is quite an interesting approach. Out of curiosity, would WITHIN fly normalisation or WITHIN fly engineering of features not allow the authors to perform a classification task like that used with the ACROSS individual analysis? e.g. normalise values within a fly, instead of across the cohort. A couple of sentences on this might just clear this up for readers.

There are various methods for batch correction which could improve the generalisability of the ACROSS top features to the independent data set. Is this something the authors have looked into? It would be good to add a sentence or two mentioning this and why it is (or isn't) applicable.

Overall, I believe the results will be of interest to the community. Moreover, I think the methodological approach is also useful, both for extending to other analyses/scientific problems and for highlighting the importance of an independent evaluation dataset.

Reviewer #3, Jacobo Diego Sitt I congratulate the authors for this very interesting work. The approach proposed is very interesting and will add an enormous exploration of time-series features for the study of consciousness.

My only methodological question refers to the FDR correction for multiple comparisons. Do the authors propose to apply multiple FDR corrections (across channels) for each feature independently or a unique FDR correction across features x channels? In my view given the number of comparisons, the latter is here the correct approach.

I would like to mention that the authors indicate that prediction generalization using independent datasets is limited in human data. I have to disagree with this statement. It is true that the first studies of biomarkers of consciousness relied on unique datasets to make predictions, but the current standard in the field is to use independent datasets to validate the proposed biomarkers, for example, see reference 14 in your manuscript for EEG or Demertzi Brain 2015 and Sci Adv 2019 for fMRI biomarkers.

Regarding the interpretation of the potential results, I would appreciate it if in the next stage the authors could address in more detail the potential limitations of the weaker form of generalization. In addition, it would be interesting to read the view of the authors of how the proposed approach fits the current discussion of the need for interpretability and explainability for machine learning / biomarkers for putative medical applications.

Reviewer #4, Tristan Bekinschtein: The aim to separate, in a data-driven manner, wakefulness (the wake state, either active or passive) from anaesthesia (the anaesthetised state) with neural data is, in principle, useful to understand the underlying signatures of each state and ultimately to further our understanding of consciousness from a neuroscientific perspective. I agree of the strength of using a completely new dataset to test once trained in the original dataset as the independence would allow to trust the features obtained and imply generalization. I am more worried about the low data amount compared to the level of features and how is that conceptualised. Finally I am asking for further interpretation at the neuroscience level once the data analyses is performed, to conceptually validate the results, as I believe that data science alone cannot define or interpret the result. I comment in more details in the following paragraphs.

I am assuming that the statistical framework, including the permutation and relationship between epochs, electrodes, conditions and flies (test and training sets) are state of the art, I cannot unfortunately be the right expert for that, although I can comment and hope for an explanation as to whether is it correct and valid to evaluate ~700 features in a small set of data (12 flies, 15 electrodes and eight epochs). This is not a critique but a plea for an explanation on how to trust the results with such limited data resources. In particular I don't understand how is FDR applied in each channel for those ~7000 features and whether is gives you a specific cut-off that you trust.

If I understand correctly for the second analysis you give a direction of effect based on the median

"Because the direction of the effect of anesthesia is not necessarily the same across features and channels, we first assigned directionality labels based on the median wakeful and anesthesia values in the 13 discovery flies. For a given feature and channel, we gave a label of 1 if the median wakeful value was greater than for anesthesia, and -1 otherwise. We then multiplied feature values by these labels, flipping the direction of the effect of anesthesia when the median wakeful value is lesser than the median anesthesia value and making the analysis uniform across features and channels. Finally, we report the average proportion across all wakeful epochs and flies."

I wonder if removing the strictly binary label of 1 and -1 and leaving the normalized difference would give you a direction and also a normalized strength of effect that would shield the results less jerky that the binarization and add more nuanced and strength to the classifiers.

Further to this, it would be good from the beginning to described some of the features, or families of features that separate between conscious states in the original data sets and generalised to the 2 flies in the pilot analyses. When having this amount of features it is possible to colour or cluster features according to its characteristics or common mathematical effects, roughly saying what they measure. I see in the manuscript the examples of specific features, but I would like a paragraph on whether there is clustering on features (for figs 2c and 3c) and which of them, as I see in the colourful patterns in the other subfigures (2a and 3a).

Can we trust this 2 features singled and maximal in figure 3c? isn't statistically fishy to have ~95% classification? Depending on the model used there is sometimes penalizing criteria to avoid overfitting classification. I wonder how the authors protect themselves from this, or if they have different interpretation. This specific result also helps me ask for clusters, and how in that see of blue dots some specific neural features families are together and how that helps interpret the results beyond the simple data-driven aspect. This plea of for us to understand if once the results are given (for the discovery and the pilot) we can go back to neural science and away from data science and start to trust the data output as meaningful for the data we have (LFP) and its conditions (awake and anaesthetised).

This aim is clearly stated in the current RR and justified but I am unsure, while reading the introduction, how is this an RR since the preliminary results are already informing. I think there is the need to state clearly which questions and analyses belong to the RR early on in the manuscript by mentioning that are different flies and how will analyses are blinded.

A detail in the intro, it might be good to defined what the authors mean by a univariate feature as for example, they ascribed non-univariate states to Lempel-Ziv Complexity and I would imagine that since it can be calculate on a single electrode, it is not a multivariate feature, or bivariate feature. This might seem a detail but it would be good to hear the classification of the feature the authors have in mind and what are the inclusion or exclusion criteria for them.

In Page 17 "We obtained random-classification distributions for the discovery and pilot evaluation flies by repeatedly classifying discovery or evaluation epochs randomly, with equal probability (as there are 7702 potentially available features in hctsa, we repeated this random classification N = 7702 times to build the distribution)." The numbers of features can be corrected to the expected number taking into consideration the limitations described in "Not all of the available time-series features could be extracted successfully from our datasets. For example, the class of features derived from the hctsa function DN_CompareKSFit includes fits of the data to a beta distribution, which assumes values between 0 and 1, an assumption that is not fulfilled by our data and consequently returns missing (NaN) values. To filter out these cases, we excluded any feature which returned NaN across all time series for a given channel in the discovery flies. This reduced the set of features down to an average of 6860 features across the 15 channels (ranging from 6657 to 7004). We further excluded features which returned a constant value across all time series for a given channel in the discovery flies because they are uninformative for classification, reducing the set of features again to on average 6764 features across the 15 channels (ranging from 6560 to 6908)."

---

## [Decision Letter · Decision Letter 2]

Dear Dr Leung,

Thank you for your patience while we considered your revised manuscript "Towards blinded classification of levels of consciousness: distinguishing wakefulness from general anesthesia and sleep in flies using a massive library of univariate time series analyses" for consideration as a Preregistered Research Article at PLOS Biology. Your revised study has now been evaluated by the PLOS Biology editors, the two Academic Editors and the original reviewers.

In light of the reviews, which you will find at the end of this email, we are pleased to offer you the opportunity to address the remaining concerns raised by Reviewer 1 in a revision that we anticipate should not take you very long. We will then assess your revised manuscript and your response to the reviewers' comments and consult with Reviewer 1 to determine whether they are satisfied with these final revisions.

**IMPORTANT - SUBMITTING YOUR REVISION**

Your revisions should address the specific points made by each reviewer. Please also submit the following files, and ensure you address the along with your revised manuscript:

*Resubmission Checklist*

*Published Peer Review*

*PLOS Data Policy*

Please note that as a condition of publication PLOS' data policy (http://journals.plos.org/plosbiology/s/data-availability) requires that you make available all data used to draw the conclusions arrived at in your manuscript. If you have not already done so, you must include any data used in your manuscript either in appropriate repositories, within the body of the manuscript, or as supporting information (N.B. this includes any numerical values that were used to generate graphs, histograms etc.).

Regardless of the method selected, please ensure that you provide the individual numerical values that underlie the summary data displayed in any graphs/heat maps as they are essential for reviewers and readers to assess your work.

Please also ensure that FIGURE LEGENDS in your manuscript include information on where the underlying data can be found, and ensure your supplemental data file/s has a legend. (This step is often forgotten!!!)

Finally, please ensure that your Data Statement in the submission system accurately describes where your data can be found.

For an example see here: http://www.plosbiology.org/article/info%3Adoi%2F10.1371%2Fjournal.pbio.1001908#s5

*Blot and Gel Data Policy*

Sincerely,

Kris

Kris Dickson, Ph.D. (she/her)

Neurosciences Senior Editor/Section Manager

PLOS Biology

kdickson@plos.org

REVIEWS:

Reviewer's Responses to Questions

Do you want your identity to be public for this peer review?

Reviewer #1: No

Reviewer #2: No

Reviewer #3: Yes: Jacobo Diego SITT

Reviewer #1: We appreciate the authors' genuine effort and the substantial improvements in the manuscript that have made since last review. It is recognized that collecting a robust data set within various readily available sources would enable the authors to draw more convincing conclusions. The inclusion of a wake/sleep data set thus could potentially improve the impact of the paper, provided that the experimental conditions are well defined and the sample size is sufficient for strong inferences. We have a few concerns and suggestions to further strengthen this study:

I. How is sleep defined in the data set?

Since the recording was performed on mounted flies, sleep indicators based on conventional locomotion activity criteria will be very limited and need additional detectable indicators. This should be spelled out explicitly.

Recent work by the Van Swinderen and Allada Labs (van Alphen et al., 2021Science Advances) has indicated different sleep states in flies correlated with proboscis pumping. Does the data set have enough flies to resolve differences in sleep state? A clear-cut result would be a desirable contribution to the field.

II. Depth of anesthesia.

Adding the new data set for a different concentration may reveal the commonality of anesthetic effect and enable a comparison to see the overlap and distinction between sleep (states) and depth of anesthesia. It is still a severe limitation in the scientific inference based on ac single drug to draw conclusions on "anesthesia" effects on LFP features. If appropriately treated, this study may become a significant contribution to the field. The authors should strive for clearly accessible results based on high-quality data sets from more than one anesthetic compound.

III. Are the findings of HCTSA parameters sex-specific?

Given the widespread recognition of sex-based differences, the authors should report the sex of the recorded flies, and if possible describe any sex-based differences found in recordings from males and females under wake/sleep/anesthetic states. It is well known that male and female flies display different circadian sleep-wake cycle patterns. An analysis of sex differences would be practically useful to others in the field, and could help identify issues/artifacts originating from electrode positioning (female fly heads are a bit larger and have more facets in their compound eyes).

IV. What is meant by "conscious levels"

A clear operational definition of "consciousness" is required. What is meant by "conscious arousal"? It can be argued that the work is more so an investigation into different sorts of "un-consciousness" and it seems that "brain states" or "arousal states" would be more appropriate terms in describing how sleep and anesthesia induce changes in LFP features associated with different categories of "un-consciousness". The title of the manuscript needs to be appropriately modified to reflect this point.

V. Number of independent samples, feature robustness, reproducibility, calibration of analytical tools

We are still concerned that the number of independent LFP recordings from different flies is too low to draw the intended conclusions. It is strongly encouraged to include additional data sets (new recordings or previously published) to enhance the validity of the conclusions. This is well within the reach of this established group with recognized expertise in this area.

This will help the authors fulfill a major goal: to examine and estimate how robust each (or at least some) of the various features extracted from the signals (time series) in the data set to identify most useful indicators:

which ones are more tolerant to (or more sensitive to) biological or experiment variability, i.e. which ones stay invariant across different biological samples and/or experimental conditions (such as electrode track precision; biological variation among individual organisms).

We were surprised to see that the original description of the multi-electrode approach was not cited (Paulk et al., 2013, J Neurophysiol). The authors should also consider using raw data of recordings from this work to improve the robustness of the analysis. Additionally, Paulk et al have a description of the variability of the electrode insertion tracks. The authors should discuss their HCTSA findings with respect to the variable electrode insertion described in Paulk.

Instead of a single epoch, analyzing additional epochs or time segments from each individual can better demonstrate certain time-invariant stationary features, uncovering useful time series features and robust parameters.

Some simple treatments that are more accessible to the general readership of PLoS Biol may be first presented and examined before attempting to interpret overall pictures with a large number of time series features. For example, in Fig R1.3, it could be readily seen whether Autocorrelation features can distinguished sleep, awake and anesthesia states if each of them display best autocorrelation values with distinct ranges of time lags.

Many of these points can be empirically determined by the authors with suitable analysis and additional data sets. Comparing the current with the new data sets can also demonstrate the reproducibility and strengthen the conclusions.

Reviewer #2:

The addition of more evaluation flies + a more detailed description of the spatial effects has boosted the proposed analysis.

Personally, the capability to accurately predict/distinguish is what interests me. However, I agree with Reviewer #1, in that whilst I can see what biological insights might come from the analyses, e.g. spatial locations associated with conciousness, I do find it difficult to see how any mechanism could be inferred from some identified time-series feature. For example, if we find a particular time-series feature that is highly predictive, how can this be linked to an underlying neurological mechanism? Of course, this extends more broadly to even traditional neuroscientific features (such as band power), and as such, I don't think its reason to hold this analysis back. Interpretation of the top predictive hctsa features and how they could be generated would lead to interesting insights into underlying mechanisms.

I have no further revision requests.

Reviewer #3: All my comments are properly addressed.

---

## [Decision Letter · Decision Letter 3]

Dear Dr Leung,

Thank you for your patience while your Pre-Registered Research Article "Towards blinded classification of loss of consciousness: distinguishing wakefulness from general anesthesia and sleep in flies using a massive library of univariate time series analyses" was re-reviewed for PLOS Biology. Your Stage 1 manuscript has been evaluated by the PLOS Biology editors, two Academic Editor with relevant expertise, and by Reviewer 1 in this last round.

Reviewer 1 is now also happy with your Stage 1 Protocol. All of the reviewers now satisfied that your Stage 1 Protocol meets our criteria for importance of research question and technical soundness of the study proposal. We are thus happy to issue a Stage 1 'in-principle acceptance' decision, with a commitment to publish the final Stage 2 Preregistered Research Article (after revision, if needed), pending successful completion of the study.

**Please carefully read all the following information. There are steps below that are required to complete to begin Stage 2.**

The study should now be completed according to the Stage 1 approved methods and analytic procedures, and the final manuscript should include an evidence-based interpretation of the results. Please see the review criteria for Stage 2 manuscripts here:

https://journals.plos.org/plosbiology/s/reviewer-guidelines#loc-reviewing-preregistered-research-articles

Subsequent editorial decisions for this study will not be based on the perceived importance or novelty of the results obtained during the data gathering and analysis phase of the work. It is critical however that you adhere to the approved Stage 1 study design when performing the study. Any deviation from these experimental procedures would need to be justified and approved by the editors (and potentially the reviewers), as otherwise it could lead to rejection of the manuscript at Stage 2. Please consult the editors immediately for advice if you need to alter this approved study plan.

**IMPORTANT**: Please follow the link below for important information regarding the Stage 2 manuscript template and review criteria. Please carefully read the guidelines on Stage 2 data collection BEFORE performing your study and completing your Stage 2 manuscript.

AUTHOR GUIDELINES: https://genweb.plos.org/Marketing/Biology%20Preregistered%20Articles%20Guidelines%20for%20Authors.pdf

*Depositing this Stage 1 Protocol*

PLOS Biology does not publish Stage 1 Protocols immediately following an in-principle acceptance. Instead they are held and integrated into a single, completed 'Preregistered Research Article' following review and acceptance of the final Stage 2 manuscript. You are however required to register this approved Stage 1 Protocol with the Center for Open Science (https://cos.io/prereg/) or another recognised repository. This may be done publicly or under private embargo until submission of the Stage 2 manuscript. Stage 1 Protocols can be quickly and easily registered using a tailored mechanism for Registered Reports (https://osf.io/rr/). Please do this now. You will need to include the URL to this deposited protocol in your Stage 2 manuscript.

*Timeline*

We understand that carrying out the study will require a significant length of time and are willing to allow you [***1 year - EDITOR TO EDIT BASED ON THE PROPOSED TIMELINE PROVIDED IN THE STAGE 1 MS. MAKE SURE TO UPDATE THE EM REVISION DATE BEFORE SENDING THIS LETTER**] to perform the study. Please email us at plosbiology@plos.org to discuss this if you have any questions or concerns, or to discuss an alternate timeline.

At this stage, your manuscript remains formally under active consideration at our journal. Please notify us by email if you do not wish to submit a Stage 2 manuscript or wish to pursue publication elsewhere, so that we may withdraw your manuscript.

*Resubmission Checklist*

Before submitting the Stage 2 manuscript, please review the following resubmission checklist: https://plos.io/Biology_Checklist

Please note that for PRA stage 2, the response to reviewers file does not follow the standard format, but should rather be a document for the reviewers detailing the changes made to the manuscript since the stage 1 accept.

*Published Peer Review*

*PLOS Data Policy*

Please note that as a condition of publication, PLOS' data policy (http://journals.plos.org/plosbiology/s/data-availability) requires that you make available all data used to draw the conclusions arrived at in your manuscript. Please note that for this article type, the raw data itself should be archived and made freely available in a public repository rather than submitted as supplementary material. Please make sure to read the Stage 2 submission guidelines online regarding how this data should be annotated and appropriately time stamped to show that data was collected after this Stage 1 in-principle acceptance and not before.

*Blot and Gel Data Policy*

To enhance the reproducibility of your results, we recommend that, if applicable, you deposit your laboratory protocols in protocols.io, where a protocol can be assigned its own identifier (DOI) such that it can be cited independently in the future. For instructions see: https://journals.plos.org/plosbiology/s/submission-guidelines#loc-materials-and-methods

Thank you again for your submission to PLOS Biology. We hope that our editorial process has been constructive thus far, and we welcome your feedback at any time. Please don't hesitate to contact us if you have any questions or comments.

Sincerely,

Kris

Kris Dickson, Ph.D. (she/her)

Neurosciences Senior Editor/Section Manager

PLOS Biology

kdickson@plos.org

REVIEWS:

Reviewer #1: The authors have carefully and thoughtfully responded to the comments, and the issues raised in the previous review are either satisfactorily resolved or will be treated in followed-up studies. The critical information in experimental methods, data analysis, and result interpretations have now been properly revised or explicitly qualified, which greatly improved the precision and rigor of presentation. As proposed, a comprehensive analysis can potentially fill in some important gaps in our understanding of the functional manifestation of different "conscious levels" and as such, a thorough treatment may identify novel time series parameters to become a valuable contribution to the literature.

---

## [Decision Letter · Decision Letter 4]

Dear Angus,

Thank you for your patience while we considered your revised manuscript "Towards blinded classification of loss of consciousness: distinguishing wakefulness from general anesthesia and sleep in flies using a massive library of univariate time series analyses" for publication as a Preregistered Research Article at PLOS Biology. Apologies for the long delay in getting back to you. As I mentioned previously, we had unexpected difficulties in finding suitable reviewers as not all of the original reviewers of your Stage 1 submission were available to re-review the revised manuscript. In any case, your revised manuscript has been evaluated by the PLOS Biology editors, the academic editors and two of the original reviewers.

Based on the reviews and on our academic editors' assessment of your revision, we are likely to accept this manuscript for publication, provided you satisfactorily address the remaining points raised by the reviewers and one comment from one of the academic editors. Please also make sure to address the following data and other policy-related requests:

* We would like to suggest a different title to improve its accessibility for our broad audience:

Wakefulness can be distinguished from general anesthesia and sleep in flies using a massive library of univariate time series analyses

* Please add the links to the funding agencies in the Financial Disclosure statement in the manuscript details.

* DATA POLICY:

Regardless of the method selected, please ensure that you provide the individual numerical values that underlie the summary data displayed in the following figure panels as they are essential for readers to assess your analysis and to reproduce it: 1CD and S3ABC.

* CODE POLICY

* Because you looked for generalization across different states (anesthesia/sleep), the underlying assumption is that anesthesia and sleep are caused by the same changes in the neural circuit, as compared to wakefulness. This is a very strong assumption and probably causes a lack of generalization of many univariate time series measures. Some measures may generalize from the discovery set to the test set for anesthesia but not for sleep, for example. This issue should be addressed more in the discussion section of the manuscript.

We expect to receive your revised manuscript within two weeks.

*Published Peer Review History*

*Press*

Sincerely,

Christian

Christian Schnell, PhD

Senior Editor

cschnell@plos.org

PLOS Biology

Reviewer remarks:

Reviewer #3: In this Stage 2 version of the manuscript, the authors present their final version of the study.

Overall, the new version follows the Stage 1 rationale, and the authors adhered precisely to the approved Stage 1 experimental procedure. The modifications highlighted in red indicate areas where the authors have updated their methods, but these changes do not fundamentally alter the original hypotheses or approach. The newly analyzed data (additional flies) robustly test the original hypotheses. However, some potential experimental differences—such as acquisition conditions or preprocessing procedures—exist between the Stage 1 and Stage 2 datasets, leading to shifts in data distribution that the authors report but had not fully understood.

In general, the discussion is supported by the results provided in the paper. However, I have two important comments that the authors need to address:

1) In line 959, the authors state: "Across-subject classification using single, specific features—which is ideal for a consciousness measure—is rarely performed, especially in the human neuroimaging literature. In the rare case where it is carried out, significant classification performance is not obtained, likely because only traditional spectral power or complexity measures are used."

I strongly disagree with this argument. Cross-validation and independent validation of neuroimaging biomarkers across subjects using single (and multivariate) features is a well-established standard in the field, particularly in clinical studies. For example, reference 14 in the manuscript employs a series of univariate features, which are cross-validated in one dataset and generalized to two independent validation datasets. This methodological framework has become a standard approach in the field. The authors should either revise their claim or provide substantial evidence supporting their position.

2) The authors do not sufficiently discuss the limitations of using sleep as a model for unconsciousness. Recent research demonstrates that human subjects can respond to commands across different sleep stages, which challenges the validity of sleep as a complete model of unconsciousness (Türker et al., Nature Neuroscience, 2024). In light of this, the reported lack of generalization across all sets of evaluation flies should be discussed within the framework of using sleep as an equally valid or limited model of unconsciousness. The authors should expand their discussion to address this nuance.

Finally, the data availability statement complies with PLOS' requirements. The authors have made preprocessed data publicly accessible in repositories such as OSF and Figshare, with appropriate timestamps confirming data collection after Stage 1 approval.

Addressing these concerns will improve the manuscript's clarity and alignment with current standards in the field.

Reviewer #4 (Tristan Bekinschtein): After the changes to the pre-registered report, the further changes and clarification during the analyses submission process I think this is ready for the consideration of the editor, I as a reviewer was happy throughout the process and engage in scientific conversation at every stage. Thanks.

---

## [Editor Report · Decision Letter 5]

Dear Angus,

Thank you for the submission of your revised Preregistered Research Article "Wakefulness can be distinguished from general anesthesia and sleep in flies using a massive library of univariate time series analyses" for publication in PLOS Biology. On behalf of my colleagues and the academic editors, Christopher Chambers and Simon van Gaal, I am pleased to say that we can in principle accept your manuscript for publication, provided you address any remaining formatting and reporting issues. These will be detailed in an email you should receive within 2-3 business days from our colleagues in the journal operations team; no action is required from you until then. Please note that we will not be able to formally accept your manuscript and schedule it for publication until you have completed any requested changes.

While you attend to those requests to come, please also make sure to mention in the legend of Figure 1 where the source data can be found.

PRESS

Sincerely, 

Christian

Christian Schnell, PhD

Senior Editor

PLOS Biology

cschnell@plos.org